# Circuit mechanisms encoding odors and driving aging-associated behavioral declines in *Caenorhabditis elegans*

**Sarah G Leinwand[1,2], Claire J Yang[2], Daphne Bazopoulou[3], Nikos Chronis[3], Jagan Srinivasan[4], Sreekanth H Chalasani[1,2]\***

[1]Neurosciences Graduate Program, University of California, San Diego, La Jolla, United States; [2]Molecular Neurobiology Laboratory, Salk Institute for Biological Studies, La Jolla, United States; [3]Department of Mechanical Engineering, University of Michigan, Ann Arbor, United States; [4]Department of Biology and Biotechnology, Worcester Polytechnic Institute, Worcester, United States

**Abstract** Chemosensory neurons extract information about chemical cues from the environment. How is the activity in these sensory neurons transformed into behavior? Using *Caenorhabditis elegans*, we map a novel sensory neuron circuit motif that encodes odor concentration. Primary neurons, AWC[ON] and AWA, directly detect the food odor benzaldehyde (BZ) and release insulin-like peptides and acetylcholine, respectively, which are required for odor-evoked responses in secondary neurons, ASEL and AWB. Consistently, both primary and secondary neurons are required for BZ attraction. Unexpectedly, this combinatorial code is altered in aged animals: odor-evoked activity in secondary, but not primary, olfactory neurons is reduced. Moreover, experimental manipulations increasing neurotransmission from primary neurons rescues aging-associated neuronal deficits. Finally, we correlate the odor responsiveness of aged animals with their lifespan. Together, these results show how odors are encoded by primary and secondary neurons and suggest reduced neurotransmission as a novel mechanism driving aging-associated sensory neural activity and behavioral declines.

**\*For correspondence:**
schalasani@salk.edu

**Competing interests:** The authors declare that no competing interests exist.

## Introduction

Animals have evolved specialized sensory systems to detect relevant information in their environment. This sensory information is relayed to downstream neural circuitry, generating appropriate food-seeking and toxin-avoiding behaviors, which enhance animal fitness. In particular, olfactory sensory neurons have an additional challenge in detecting a large set of volatile cues (*Buck, 2005*). In mammals, odors are detected by G-protein coupled odorant receptors that are expressed on olfactory sensory neurons. Moreover, while the mammalian genome encodes approximately 1000 receptors (*Buck and Axel, 1991*), each olfactory sensory neuron is known to express only one type of receptor (*Vassar et al., 1993*; *Chess et al., 1994*). Since mammals can detect far more than 1000 odors (*Duchamp-Viret et al., 1999*; *Rubin and Katz, 1999*), this suggests that olfactory information at this level is encoded by a combinatorial code (*Malnic et al., 1999*). Calcium imaging and electrophysiological studies have confirmed that individual odorants bind multiple odorant receptors and activate the corresponding olfactory sensory neurons (*Malnic et al., 1999*; *Abaffy et al., 2006*). Moreover, activity in an individual olfactory sensory neuron represents not only the molecular receptive field of its odor receptors (*Araneda et al., 2000*), but also gating by feedback circuits (*Gomez et al., 2005*; *Wachowiak et al., 2009*) and modulation by sniffing behavior in mammals (*Wesson et al., 2009*). Information from these sensory neurons is then further processed and relayed to other brain regions (*Ghosh et al., 2011*;

**eLife digest** A sense of smell can help animals to find food and detect danger. Odor molecules activate so-called olfactory neurons that relay signals to the brain in the form of nerve impulses. This information is then processed, and the appropriate response is triggered; for example, an animal might move towards the smell of food, or away from the scent of a predator. But how can the activity of olfactory neurons trigger the right behavioral response?

Leinwand et al. have now explored the activity of olfactory neurons in a roundworm called *C. elegans*. The experiments revealed that a food odor activated two olfactory neurons directly, and that each of these 'primary' neurons then in turn activated another 'secondary' olfactory neuron. This communication between primary and secondary olfactory neurons was essential for worms to respond to the food odor. Further experiments revealed that the primary olfactory neurons send chemical signals, called neurotransmitters and neuropeptides, to communicate with the secondary neurons. Importantly, mutations that blocked this chemical signaling prevented the worms from responding appropriately to the smell of food.

Aging animals, including people, often have impaired senses and can therefore find it difficult to identify and respond to odors. Leinwand et al. found that aged worms were no different. Further experiments suggested that aging worms' responses to odor decline because the communication between the primary and secondary olfactory neurons may be impaired with age. When Leinwand et al. strengthened this communication it reversed the effects of aging on the worms' sense of smell. Moreover, the experiments also showed that an animal's performance on the odor task was correlated with its longevity, such that the better performers also lived longer. A challenge for the future is to understand the precise changes that occur at early stages of aging to impair the sense of smell. Future studies could also test if similar combinations of olfactory neurons are needed to trigger certain behavioral responses to odors in young and old mammals.

*Miyamichi et al., 2011*; *Sosulski et al., 2011*). Despite this understanding, little is known about how specific activity patterns in the olfactory sensory neurons are correlated with behavioral outputs. One solution to this problem is to analyze numerically simpler invertebrate olfactory circuits where information flow can be traced at the resolution of individual neurons and correlated with animal behavior.

The nematode *Caenorhabditis elegans*, with its small nervous system consisting of just 302 neurons, is ideally suited for a circuit-level analysis of chemosensory processing and behavior. Chemosensory stimuli are detected by twelve sensory neuron pairs located in the amphid ganglia (*Figure 1A*) (*White et al., 1986*; *Bargmann, 2006*). All 24 of these neurons send their dendrites to the nose of the animal where they detect environmental changes and relay that information through their axons to the downstream circuitry (*White et al., 1986*). *C. elegans* uses small numbers of sensory neurons to drive locomotion towards or away from particular sensory stimuli (*Bargmann, 2006*). For example, single cell ablation experiments showed that the bilaterally asymmetric pair of AWC sensory neurons is necessary for attraction to benzaldehyde (BZ) odor, while the AWA sensory neuron pair is required for diacetyl odor attraction (*Bargmann and Horvitz, 1991*; *Bargmann, 2006*). Functional imaging experiments revealed that AWC neurons are activated by the removal of odor stimuli (*Chalasani et al., 2007*), while AWA neurons respond to the addition of odors (*Zaslaver et al., 2015*). However, these sensory neuron activity patterns are not sufficient to explain how animals behave when they encounter diverse olfactory stimuli in the environment. We hypothesized that multiple amphid ganglia neurons could encode odor information and drive plastic olfactory behaviors; therefore, we performed the first comprehensive analysis of odor-evoked neural activity in all amphid neurons. We identified a novel circuit motif consisting of primary and secondary olfactory neurons that collectively encode odor and drive behavioral plasticity. We then analyzed the reliability of this combinatorial code and found that it degrades during aging. Our experiments suggest that a selective vulnerability of neurotransmitter release pathways in aged animals is the underlying mechanism that leads to a specific decay in secondary olfactory neuron activity and associated behavioral decline. Furthermore, we find that olfactory circuit function is correlated with an animal's longevity.

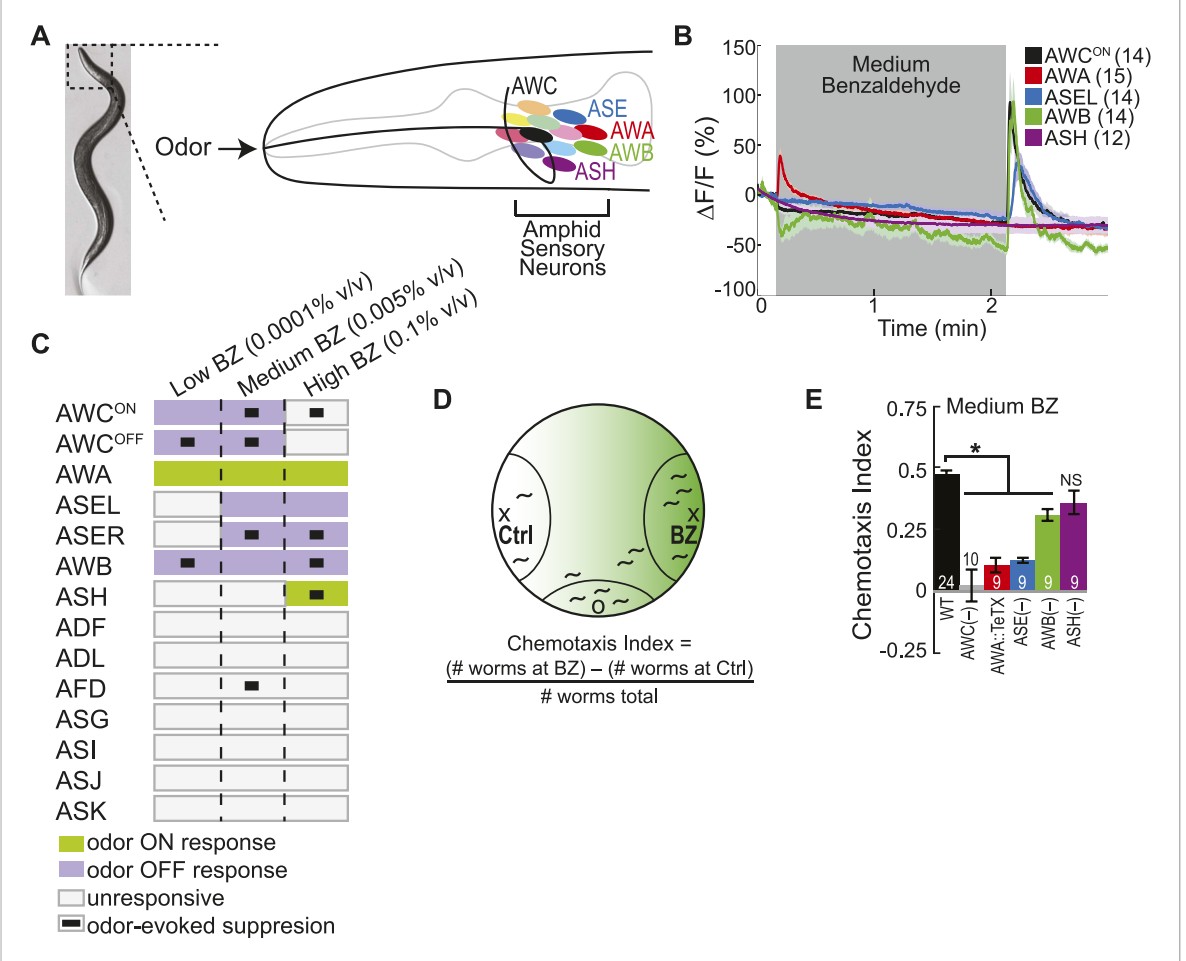

Figure 1. Multiple sensory neurons detect the odor benzaldehyde (BZ). (A) Image of a young adult *C. elegans* and schematic depicting the twelve pairs of sensory neurons in the anterior amphid ganglia whose dendrites project to the nose of the animal where they detect sensory stimuli. (B) Average GCaMP fluorescence change in young adult (day 1), wild-type sensory neurons in response to medium concentration (0.005% vol/vol) BZ stimulation. Shaded box indicates two minute BZ odor stimulation beginning at t = 10 s. The light color shading around curves indicates s.e.m. and numbers in parentheses indicate number of neurons imaged. (C) Summary chart of the calcium responses of all amphid sensory neurons to low (0.0001% vol/vol), medium (0.005% vol/vol) and high (0.1% vol/vol) concentrations of BZ odor. This chart shows the composition of the *C. elegans* olfactory neural circuit and depicts a combinatorial sensory neuron code for odor concentration. The calcium signal in some neurons (as indicated) is suppressed by the addition of odor (see methods and materials section). (D) Chemotaxis assay schematic depicting *C. elegans* attraction to a point source of BZ. Animals are placed at the origin (O) and allow to chemotax towards a point of BZ or control (Ctrl). The putative BZ gradient is shown in shades of green with darker colors representing higher BZ concentrations. (E) Young adult (day 1) chemotaxis performance of wild-type, AWC or AWB or ASH neuron-specific genetic ablation, AWA neuron-specific tetanus toxin expression worms or *che-1* mutants missing ASE neurons to a medium concentration point source of BZ odor (*Uchida et al., 2003*). See *Figure 1—source data 1* for raw chemotaxis data. Numbers on bars indicate number of assay plates and error bars indicate s.e.m. *p < 0.05, two-tailed *t*-test with Bonferroni correction, compared to wild-type.

The following source data and figure supplement are available for figure 1:

**Source data 1**. Young adult chemotaxis performance data.

**Source data 2**. Odor-evoked responses in wild-type young adult data.

**Figure supplement 1**. Combinatorial olfactory coding in *C. elegans*.

## Results

### Multiple sensory neurons detect the food odor Benzaldehyde (BZ)

We used functional imaging to identify the amphid sensory neurons that detect the food odor Benzaldehyde (BZ) (*Figure 1A*). We trapped young adult animals expressing GCaMP family of genetically encoded calcium indicators (*Tian et al., 2009*), under cell selective promoters, in individual amphid sensory neurons in our custom-designed microfluidic device (*Chalasani et al., 2007*) and recorded their responses to BZ. Consistent with previous studies, we observed a large calcium transient indicating increased AWC activity upon removal of a medium concentration BZ stimulus (*Figure 1B,C*, *Figure 1—figure supplement 1A,B*) (*Chalasani et al., 2007*). Unexpectedly, we found additional BZ responsive neurons: the diacetyl sensing AWA neurons (*Bargmann et al., 1993*) were activated by the addition of BZ, while ASE and AWB neurons (that were previously shown to sense salts [*Bargmann and Horvitz, 1991*] and volatile repellents [*Troemel et al., 1997*; *Bargmann, 2006*], respectively) also responded to the removal of this stimulus in young adults (*Figure 1B,C*, *Figure 1—figure supplement 1A,B*). Furthermore, none of the other amphid neurons responded to this medium concentration BZ stimulus (*Figure 1B,C Figure 1—figure supplement 1C*). While the two AWC and ASE neurons can be genetically and functionally separated (*Wes and Bargmann, 2001*; *Suzuki et al., 2008*), each one in the pair showed similar responses to the removal of the BZ stimulus; therefore, we chose to focus our subsequent analysis on the AWC$^{ON}$ and ASEL (left) neurons (*Figure 1B,C*, *Figure 1—figure supplement 1D,E*). We also noted that the ASEL responses to BZ were slower to reach the maximum response (average of 5.46 s after stimulus change) than the other odor responsive neurons (average 1–2 s after stimulus change), indicating that the kinetics of odor-evoked activity are different in different cells (*Figure 1—figure supplement 1B*). Moreover, different neural activity patterns distributed across AWC, ASE, AWA, AWB and ASH sensory neurons defined active neural circuits for different concentrations (medium as well as high or low) of BZ (*Figure 1C*, *Figure 1—figure supplement 1F–I*). We focused on responses to the attractive medium concentration of BZ for the remainder of this study. Our data suggests that four pairs of sensory neurons (AWC, AWA, ASE and AWB) signal the presence of this BZ stimulus.

Next, we tested whether all four of these sensory neuron pairs were also required to drive behavioral attraction to BZ. We used a chemotaxis assay (*Figure 1D*) and analyzed the behavior of animals with non-functional sensory neurons. We found that genetic ablation (*Beverly et al., 2011*; *Yoshida et al., 2012*) or blocking synaptic transmission (with tetanus toxin [*Schiavo et al., 1992*]) in any of the four AWC, ASE, AWA or AWB neurons impaired animals' chemotaxis to a point source of medium BZ (*Figure 1E*). This data is consistent with our imaging experiments and confirms a role for multiple sensory neurons in driving attraction to the BZ odor. In particular, our results showing important roles for ASE, AWA and AWB neurons in BZ attraction are novel. Together, these results show that a combinatorial code of activity across multiple neurons is essential to drive plasticity in an animal's behavior to BZ odor.

### Primary and secondary olfactory neurons encode BZ odor

Previously, we defined two classes of sensory neurons: primary neurons, which directly detect stimuli, and secondary neurons, which respond to neurotransmission from primary neurons (*Leinwand and Chalasani, 2013*). To classify the BZ-responsive neurons, we combined laser cell ablation with functional imaging. We predicted that BZ responses in primary neurons would be preserved when other odor responsive sensory neurons were ablated, while secondary neuron responses would require functional signaling from intact primary neurons. We found that AWC$^{ON}$ responses to BZ were not affected in animals with any of the other BZ responsive neuron pairs (AWA, ASE or AWB neurons) ablated, suggesting that AWC$^{ON}$ neurons directly detect the odor stimulus (*Figure 2A*). Similarly, AWA responses to BZ were not affected in animals with ablated AWC, ASE or AWB neurons (*Figure 2B*). These experiments suggest that AWC$^{ON}$ and AWA neurons directly detect BZ and function as primary sensory neurons. In contrast, ASEL responses to BZ were greatly reduced in animals with ablated AWC neurons, but unaffected by AWA or AWB neuron ablation (*Figure 2C*). This suggests that ASEL neurons may respond to signals from AWC$^{ON}$ primary sensory neurons (*Figure 2C*). Similarly, AWB responses to BZ required signaling from AWA neurons as these responses were significantly reduced specifically in the AWA ablation condition (*Figure 2D*).

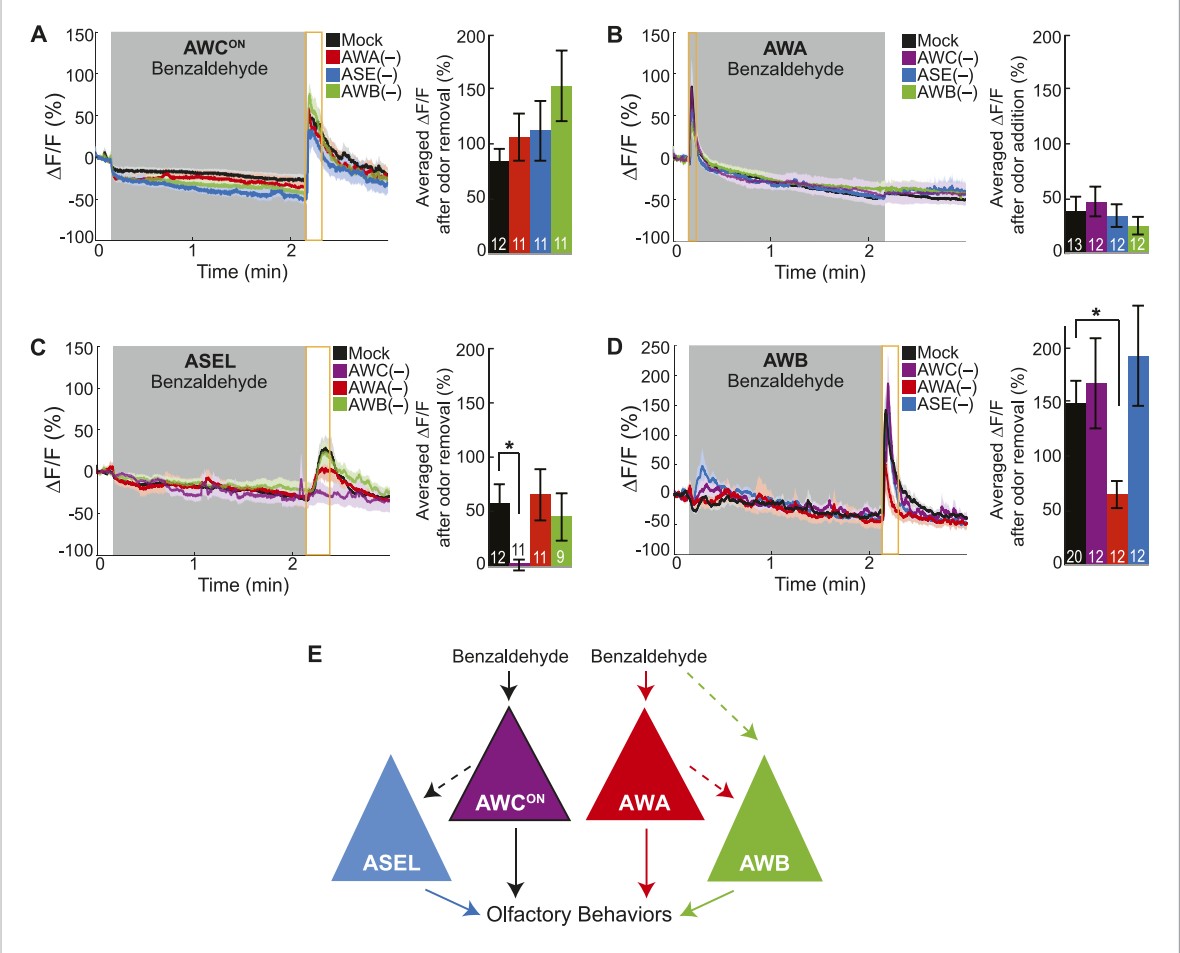

**Figure 2**. Cell ablation reveals primary and secondary BZ sensory neurons. (**A**) Average young adult AWC[ON] neuron responses to medium BZ in control (Ctrl) mock-ablated animals compared to animals with the AWA, ASE or AWB sensory neurons ablated (neurons ablated at an early larval stage). (**B**) Average young adult AWA neuron responses to BZ in Ctrl mock-ablated animals compared to animals with AWC, ASE or AWB sensory neurons ablated. (**C**) Average young adult ASEL neuron responses to BZ in Ctrl mock-ablated animals compared to animals with AWC, AWA or AWB sensory neurons ablated. (**D**) Average young adult AWB neuron responses to BZ in Ctrl mock-ablated animals compared to animals with AWC, AWA or ASE sensory neurons ablated. (**A–D**) Shaded box represents two minute medium BZ (0.005% vol/vol) stimulation beginning at t = 10 s. Yellow box indicates the time period after stimulus change for which the fluorescence change was averaged in the bar graphs (See *Figure 2—source data 1* for raw data.). Light shading around curves and bar graph error bars indicate s.e.m. Numbers on bars indicate number of neurons imaged. *p < 0.05, two-tailed *t*-test with Bonferroni correction, compared to mock-ablation. (**E**) Schematic of the BZ circuit depicting the primary, direct BZ sensory neurons and the secondary, indirect BZ sensory neurons whose odor responses are reduced by cell ablation.

The following source data is available for figure 2:

**Source data 1**. Odor responses in cell ablated animal data.

Interestingly, while AWA neurons responded to the addition of odor stimulus with an increase in the calcium signal, the AWB neuron calcium signal increased upon odor removal (*Figure 2B,D*). We suggest that AWB neurons may be inhibited by AWA and, when odor is removed, AWA is no longer active, leading to a release from inhibition and an increase in AWB activity. Additionally, direct olfactory sensory inputs or signaling from other neurons may also contribute to AWB activity, accounting for the residual AWB responses to odor in the AWA neuron ablated animals (*Figure 2D*). Collectively, these data show a novel sensory circuit configuration in which the odor responsive neurons are not equal: the olfactory circuit for BZ odor is composed of two primary sensory neurons (AWC[ON] and AWA) and two secondary neurons (ASEL and AWB) (*Figure 2E*).

## AWC-released neuropeptides and AWA-released classical neurotransmitters are required for the activity of ASEL and AWB neurons, respectively

Based on the *C. elegans* wiring diagram (*White et al., 1986*), we hypothesized that the primary olfactory neurons use chemical neurotransmission to signal the presence of odor to the secondary neurons. To identify the relevant primary neuron released neurotransmitters that activate the secondary neurons, we analyzed the neural activity patterns in various mutants. We first examined genetic mutants that primarily block the release of (1) small, clear synaptic vesicles containing classical neurotransmitters such as glutamate, gamma-aminobutyric acid (GABA) and acetylcholine [Munc13 or *unc-13* in *C. elegans* (*Richmond et al., 1999*)] or (2) neuropeptide-containing dense core vesicles [CAPS, calcium-dependent activator protein for secretion, or *unc-31* in *C. elegans* (*Speese et al., 2007*)]. We found that AWC$^{ON}$ and AWA neurons retained their odor responsiveness in the absence of classical or peptidergic neurotransmission (*Figure 3A,B*). This data confirms our cell ablation results, indicating that these neurons directly detect BZ and are primary olfactory sensory neurons. Interestingly, we found that AWC$^{OFF}$ responses to BZ were significantly reduced in *unc-13* mutants, suggesting that classical neurotransmission might be required to potentiate odor-evoked activity in this neuron (*Figure 3—figure supplement 1A*). We suggest that AWC$^{OFF}$ responses to BZ might be potentiated by classical neurotransmission from AWC$^{ON}$ neuron. Together, these results confirm that AWC$^{ON}$ and AWA are primary sensory neurons and can directly detect BZ in the environment.

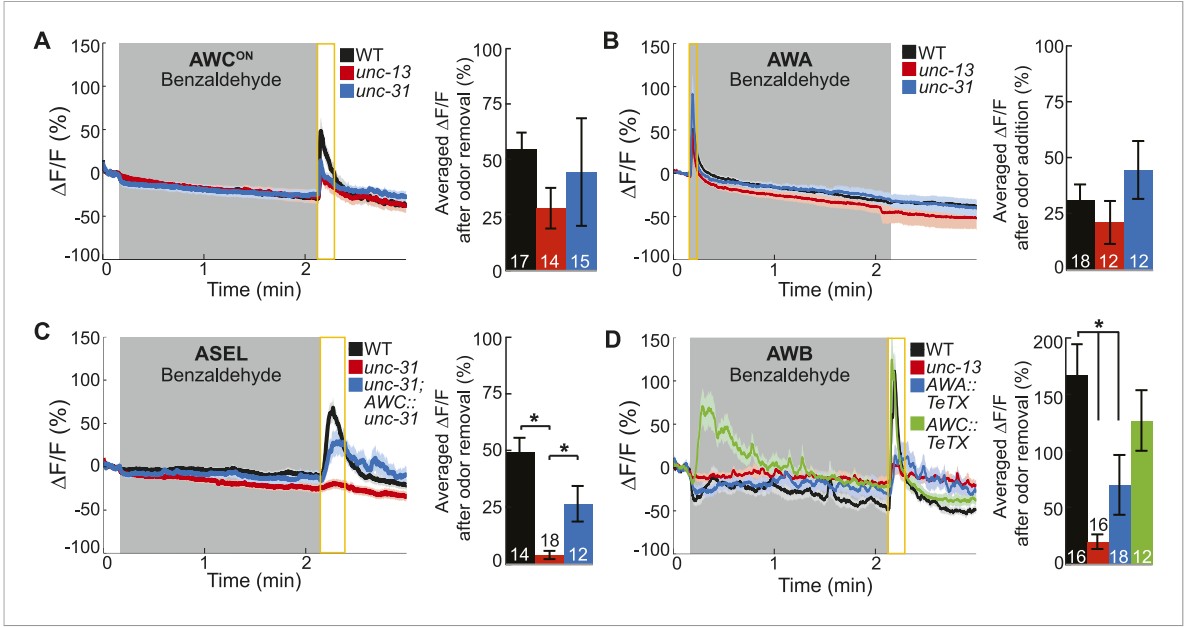

**Figure 3**. Primary olfactory neurons release neuropeptides and classical neurotransmitters to recruit secondary neurons into the BZ circuit. (**A**, **B**) Average young adult (**A**) AWC$^{ON}$ and (**B**) AWA neuron calcium responses to BZ in wild-type, *unc-13* mutants with impaired synaptic vesicle release, and *unc-31* mutants with impaired dense core vesicle release. (**C**) ASEL responses to BZ in *unc-31* mutants and *unc-31*; AWC-specific *unc-31* rescue. (**D**) AWB responses to BZ in *unc-13* mutants and animals with AWA- or AWC-specific expression of tetanus toxin. (**A–D**) Shaded box indicates two-minute medium BZ (0.005% vol/vol) odor stimulation. Yellow box indicates the time period after stimulus change for which the fluorescence change was averaged in the bar graphs (See *Figure 3—source data 1* for raw data). The light color shading around curves and bar graph error bars indicate s.e.m. Numbers on bars indicate number of neurons imaged. *p < 0.05, two-tailed *t*-test with Bonferroni correction, compared to wild-type or mutant as indicated.

The following source data and figure supplement are available for figure 3:

**Source data 1**. Odor responses in neurotransmitter release pathway genetic mutant data.

**Source data 2**. Odor responses in genetic mutant data.

**Figure supplement 1**. Primary and secondary olfactory neurons respond to BZ.

In contrast, we found that mutations impairing neurotransmission affected the odor responses of the ASEL and AWB secondary neurons, which we identified by cell ablation experiments. Specifically, odor-evoked ASEL activity required *unc-31*-dependent neuropeptide signaling (*Figure 3C*). Restoring neuropeptide release function specifically to the AWC neurons rescued ASEL BZ responses in *unc-31* mutants, suggesting that AWC neurons release peptides to recruit ASEL neurons (*Figure 3C*). This gene mutant analysis suggests that the longer time required for ASEL neurons to reach their maximum response to odor may reflect the additional requirement of AWC-dependent peptidergic transmission (*Figure 3C* and *Figure 1—figure supplement 1B*). Similarly, ASER responses to BZ also require neuropeptide signaling (*Figure 3—figure supplement 1B*). While we have not identified the source of these neuropeptides, we suggest that AWC released peptides may also activate ASER neurons. Moreover, *unc-13*-dependent classical neurotransmission was not required for either ASEL or ASER responses to BZ (*Figure 3—figure supplement 1B,C*). We then examined AWB responses to BZ in neurotransmission mutants. AWB responses were significantly and specifically reduced in *unc-13* mutants, suggesting that these neurons are recruited to this olfactory circuit by classical neurotransmitter (s) (*Figure 3D*, *Figure 3—figure supplement 1D*). To confirm that AWA was the source of these classical neurotransmitters (*Figure 2D*), we used tetanus toxin to manipulate the neurotransmitter pathways. Tetanus toxin has been previously shown to cleave synaptobrevin and block neurotransmission (*Schiavo et al., 1992*). We found that expressing tetanus toxin specifically in the AWA, but not AWC, sensory neurons significantly reduced AWB responses to BZ removal (*Figure 3D*). This confirms that AWA signals to AWB and recruits it into the odor circuit. Nevertheless, the residual odor-evoked AWB responses observed in *unc-13* mutants and transgenic animals with reduced AWA neurotransmission (AWA::tetanus toxin) confirm that direct sensory inputs or signaling from other neurons may also contribute to AWB activity (*Figure 3D*). These data show that ASE and AWB neurons can function as secondary neurons because their responses to BZ require neuropeptide and classical neurotransmitter signaling respectively. Collectively, this defines a BZ odor-encoding circuit motif consisting predominantly of two primary and two secondary neurons wired as two parallel channels of olfactory information.

## Insulin peptidergic and cholinergic transmission from primary olfactory sensory neurons are required for secondary olfactory neuron activity

We then mapped the identities of the neuropeptide and neurotransmitter pathways transferring information from primary to secondary olfactory neurons. The *C. elegans* genome includes at least 122 neuropeptide genes and pathways to generate several classical neurotransmitters including glutamate, GABA and acetylcholine (*Hobert, 2013*). To identify the cognate neuropeptide(s) activating ASEL neurons, we used ASEL activity as readout to screen a number of neuropeptide gene mutants. We found that the insulin-like peptide *ins-1* (*Pierce et al., 2001*) was required for BZ-evoked ASEL responses (*Figure 4A*). Moreover, restoring INS-1 function specifically to AWC neurons, but not to AWA neurons, rescued mutant ASEL activity deficits (*Figure 4A*). This suggests that AWC neurons release INS-1 peptides to recruit ASEL neurons into the odor circuit. To confirm AWC as the source of the INS-1 peptides, we used an AWC neuron-specific RNAi approach to knockdown the *ins-1* gene. Previous studies have shown that expressing the sense and anti-sense transcript under a cell-specific promoter can efficiently knockdown the gene of the interest in that cell (*Esposito et al., 2007*; *Leinwand and Chalasani, 2013*). We found that knocking down *ins-1* in AWC neurons significantly reduced the ASEL responses to BZ, confirming that AWC-released INS-1 is required for ASEL activity in the odor circuit (*Figure 4B*). We suggest that the same insulin neuropeptide may be multifunctional. For example, INS-1 released from AIA interneurons inhibits AWC and ASER activity (*Tomioka et al., 2006*; *Chalasani et al., 2010*), while we show that INS-1 released from AWC recruits ASEL into the BZ circuit. Ultimately, this signaling can regulate odor circuit dynamics, salt chemotaxis plasticity and integrative thermotactic behavior (*Kodama et al., 2006*; *Tomioka et al., 2006*; *Chalasani et al., 2010*). Collectively, these results suggest that the site of release and likely also signaling in the downstream neurons play key roles in determining the functionality of INS-1 peptides.

Next, we investigated the receptor and downstream signaling components in ASEL neurons that transduce the AWC-released INS-1 signal. We found that odor-evoked ASEL activity required the canonical insulin receptor (*daf-2* in *C. elegans* [*Pierce et al., 2001*]) and PI3-Kinase (*age-1* in *C. elegans* [*Morris et al., 1996*]) signaling in ASEL neurons (*Figure 4C,D*). We suggest that the increase in

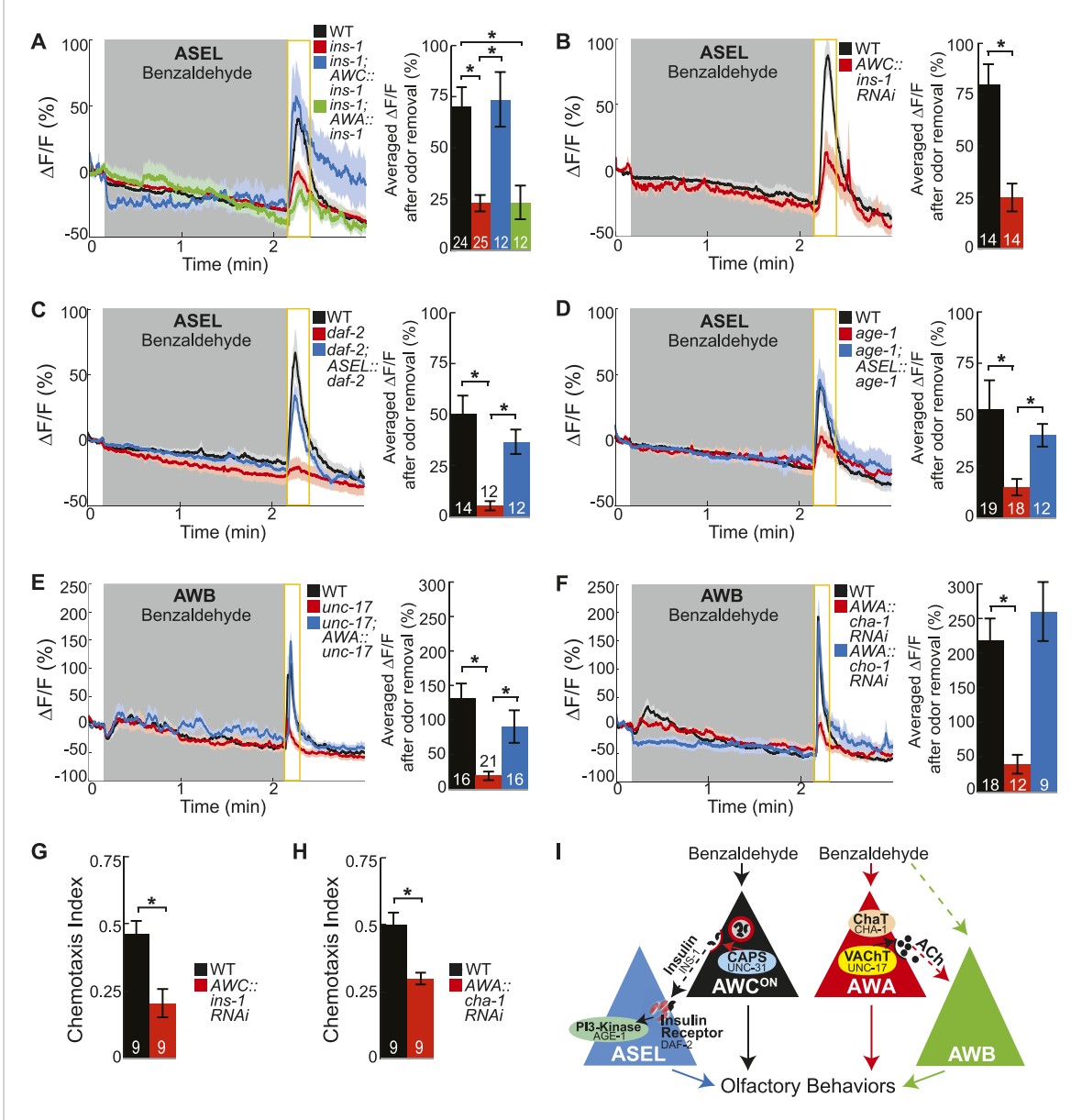

**Figure 4**. Insulin peptidergic and cholinergic transmission from the two primary olfactory sensory neurons recruits two secondary olfactory neurons. (**A**) BZ-evoked activity in young adult ASEL neurons in wild-type, *ins-1* insulin-like peptide mutants, *ins-1*; AWC-specific *ins-1* rescue and *ins-1*; AWA-specific *ins-1* rescue. (**B**) Average ASEL responses to BZ in young adult wild-type and AWC neuron-specific *ins-1* RNAi knockdown animals. (**C,D**) BZ-evoked activity in young adult ASEL neurons in (**C**) *daf-2* insulin receptor mutants and *daf-2*; ASEL-specific *daf-2* rescue, and (**D**) *age-1* PI3-Kinase mutants and *age-1*; ASEL-specific *age-1* rescue compared to wild-type. (**E**) AWB neuronal activity in response to BZ in young adult wild-type, *unc-17* vesicular acetylcholine transporter mutants and *unc-17*; AWA-specific *unc-17* rescue. (**F**) AWB neuronal activity in response to BZ in young adult wild-type, AWA neuron-specific *cha-1* choline acetyltransferase RNAi and AWA-specific *cho-1* choline transporter RNAi knockdown transgenic animals. (**G,H**) Young adult chemotaxis performance of wild-type and (**G**) AWC neuron-specific *ins-1* RNAi knockdown or (**H**) AWA neuron-specific *cha-1* RNAi knockdown animals to a medium concentration point source of BZ odor. Numbers on bars indicate number of assay plates and error bars indicate s.e.m. *p < 0.05, two-tailed *t*-test. (**I**) Proposed young adult BZ circuit model. (**A**-**F**) Shaded box indicates medium BZ (0.005% vol/vol) odor stimulation. Yellow box indicates the time period after stimulus change for which the fluorescence change was averaged in the bar graphs. Numbers on bar graphs indicate number of neurons imaged. Light color shading around curves and bar graph error bars indicate s.e.m. *p < 0.05, two-tailed *t*-test with Bonferroni correction, compared to wild-type or mutant as indicated. See also *Figure 4—source data 1* for raw data.

The following source data and figure supplement are available for figure 4:

**Source data 1**. Odor responses and chemotaxis performance in insulin and acetycholine pathway mutant and transgenic data.

*Figure 4. continued on next page*

*Figure 4. Continued*

**Source data 2**. Additional odor responses in insulin and acetycholine pathway mutant and transgenic data.

**Figure supplement 1**. Odor-evoked calcium dynamics in genetic mutants.

calcium in ASEL may result from rapid signaling downstream of PI3-Kinase acting directly on calcium channels (*Blair and Marshall, 1997*), and that this may represent an alternate pathway to the canonical, long term effects of DAF-2 signaling to regulate gene expression (*Murphy et al., 2003*). Furthermore, we found that the insulin receptor mutant (*daf-2*) had a stronger reduction in ASEL activity compared to the insulin ligand mutant (*ins-1*) or the insulin ligand knockdown animals (*Figure 4A–C*). These results suggest that AWC neurons may co-release additional insulin peptides along with INS-1 to bind the insulin receptor on ASEL neurons. Importantly, we also confirmed that AWC<sup>ON</sup> primary olfactory neuron dynamics were normal in all of the insulin pathway mutants analyzed, indicating that insulin signaling functions downstream of primary olfactory sensory transduction (*Figure 4—figure supplement 1A–C*). Furthermore, we have previously shown that ASEL responses to a different, directly detected stimulus, salt, are not affected in the *daf-2* or *age-1* mutants, suggesting that these primary ASEL responses do not depend on insulin signaling (*Leinwand and Chalasani, 2013*). Together, these results indicate that AWC-released insulin peptides signal to ASEL secondary neurons via the insulin receptor and PI3-Kinase to encode the BZ stimulus.

We also mapped the classical neurotransmitter pathway recruiting AWB neurons into the circuit. We found that mutations in the vesicular acetylcholine transporter (VAChT), *unc-17*, which packs acetylcholine into synaptic vesicles (*Alfonso et al., 1994*), reduced AWB odor responses (*Figure 4E*). Restoring cholinergic function specifically in AWA primary neurons was sufficient to elicit wild-type-like activity in AWB secondary neurons (*Figure 4E*). We also examined additional components of the cholinergic synthesis and release pathway through a cell-specific RNAi knockdown approach. We found that knocking down the *C. elegans* choline acetyltransferase (ChAT), *cha-1*, which is required for the biosynthesis of acetylcholine (*Rand and Russell, 1984*; *Alfonso et al., 1994*), specifically in the AWA neurons significantly reduced AWB neuron responses to BZ (*Figure 4F*). Together, these results suggest an essential role for cholinergic signaling from the AWA neurons to recruit AWB neurons to the olfactory circuit. Interestingly, AWA neuron-specific knockdown of the choline transporter *cho-1*, which is required for high affinity choline reuptake at presynaptic terminals (*Okuda et al., 2000*), had no effect on AWB responses to BZ (*Figure 4F*). Therefore, we suggest that AWA requires the choline acetyltransferase, but may not require the high affinity choline transporter to release acetylcholine. While we cannot rule out the possibility that our attempts to knockdown the choline transporter were ineffective, our results are consistent with prior observations that loss of *cho-1* has only mild effects on cholinergic neurotransmission and suggest that de novo choline synthesis and low affinity choline uptake may be sufficient for cholinergic signaling in the olfactory circuit (*Mullen et al., 2007*). We considered whether acetylcholine modulates AWB activity by acting on muscarinic receptors. We found that odor-evoked AWB activity was not affected in mutants of any of the three identified *C. elegans* muscarinic receptors (*gar-1*, *gar-2* and *gar-3*) (data not shown), suggesting that acetylcholine might bind other receptors on AWB neurons. The *C. elegans* genome encodes 8 acetylcholine-gated chloride channels (*Hobert, 2013*) and we suggest that AWA-released acetylcholine binds one of these receptors to inhibit AWB neuronal activity when odor is added, leading to a rebound from this inhibition when odor is removed. Moreover, we found that AWB responses to the directly detected repulsive odorant 2-nonanone (*Troemel et al., 1997*) were normal in *unc-17* mutants (*Figure 4—figure supplement 1D*). We suggest that AWB secondary (to BZ), but not primary (to 2-nonanone) responses require cholinergic signaling. Importantly, we also confirmed that AWA primary olfactory neuron dynamics were normal in the genetic mutants and knockdown animals analyzed (*Figure 4—figure supplement 1E,F*). These experiments support the conclusion that changes in the secondary neuron activity observed in these mutants and knockdown transgenic animals are downstream of sensory transduction in the primary neurons and related to transmitter release from primary neurons.

Next, we tested whether insulin peptidergic and cholinergic signaling were required for chemotaxis behavior. Consistent with our imaging results, we found that knocking down the insulin-like peptide *ins-1* in AWC neurons significantly reduced attraction to BZ (*Figure 4G*). In addition, animals with the choline

acetyltransferase *cha-1* knocked down specifically in AWA neurons also displayed significantly reduced BZ chemotaxis behavior (*Figure 4H*). Together, these data show that BZ stimulus is encoded by AWC<sup>ON</sup> and AWA primary sensory neurons, which use insulin peptidergic and cholinergic neurotransmission to elicit activity in ASEL and AWB secondary neurons and to shape chemotaxis behavior (*Figure 4I*). Thus, multiple neuropeptide and neurotransmitter pathways are integrated to shape odor encoding and behavior.

## Attractive olfactory behavior and odor-evoked activity of secondary neurons specifically decay with aging

We have shown that a combinatorial neural activity code comprising primary and secondary neurons encodes odors and drives behavior. Is this combinatorial olfactory code persistent and reliable throughout life? Interestingly, olfactory behavioral performance has been previously shown to degrade with age, which in turn affects quality of life and overall safety and survival across species (*Doty and Kamath, 2014*). We used the detailed characterization of the combinatorial BZ olfactory circuit described above to investigate systems levels changes in olfactory function with age.

We first tested whether aging affects BZ-evoked behavior. While young adults were strongly attracted to BZ odor, we found that older animals (day 4–6) showed a significant impairment in their attraction (*Figure 5A*). The behavioral deficit was largest for day 6 adults; however, we found that these animals had more variability in their size (*Figure 5—figure supplement 1A*) making it difficult to design an effective trap to image animals beyond day 5 of adulthood and analyze their odor-evoked neuronal activity. Therefore, for the remainder, we compared young adults (day 1, the age characterized above) and animals at a post-reproductive, early stage of aging (day 5), which we refer to as 'aged' adults. Importantly, we found that the aging-associated chemotaxis behavioral deficit is unlikely to be caused by changes in locomotory ability since the speed of chemotaxing aged animals did not differ from that of young adults (*Figure 5B*). This data establishes BZ chemotaxis as a model of aging-associated olfactory sensory behavioral decline.

To determine the mechanism underlying this aging-associated decline in BZ-directed behavior, we probed neuronal activity in the combinatorial, BZ-encoding sensory neural circuit described above. We analyzed the responses of the primary (AWC<sup>ON</sup> and AWA) and secondary (ASEL and AWB) neurons to BZ in both young (day 1) and aged (day 5) adult animals. Overall, aging did not affect the reliability, duration or magnitude of odor-evoked activity in AWC<sup>ON</sup> and AWA primary neurons (*Figure 5C,D,G–I*, *Figure 5—figure supplement 1B*). In contrast, odor-evoked ASEL and AWB secondary neuron activity was highly variable with aging, with many neurons failing to show any responses to odor, revealing a possible mechanism for behavioral decline (*Figure 5E–I*, *Figure 5—figure supplement 1B*). Interestingly, the AWB neurons that did respond to odor in aged animals had calcium transients that were indistinguishable from responses in younger animals (*Figure 5F–H*, *Figure 5—figure supplement 1B*). Additionally, considering only the animals with odor responsive ASEL neurons, the BZ responses of the aged animals were in fact significantly larger than that of the young animals (*Figure 5E,G,H*). These results suggest that odor-evoked activity in ASEL and AWB secondary neurons selectively decays in some animals. Consistent with these results, we found that the weak chemotaxis performance of aged animals towards BZ only required the primary AWC and AWA neurons, and not the unreliable secondary ASE and AWB neurons (*Figure 5J*). To further examine this, we tested whether performance in the chemotaxis assay is correlated with the odor responsiveness of the ASEL and AWB secondary neurons (*Figure 5—figure supplement 2A*). We found that aged animals that failed to chemotax towards BZ were significantly more likely to have odor non-responsive ASEL and AWB neurons than aged animals that successfully found the odor source (*Figure 5—figure supplement 2B–D*). Taken together, these data reveal a distributed neural circuit that detects attractive odors and suggest that BZ behavioral declines arise from unreliable activity of aged secondary ASEL and AWB neurons in this circuit.

We then tested whether this aging-associated decline was dependent on odor concentration. We showed that a distinct, but overlapping set of sensory neurons encodes high concentration BZ (*Figure 1C*, *Figure 1—figure supplement 1H,I*). Behaviorally, we found that high BZ was similarly repulsive in young and aged animals (through day 5) (*Figure 5—figure supplement 3A*). Consistently, high BZ-evoked neural activity did not significantly decline between day 1 and day 5 adults (*Figure 5—figure supplement 3B–D*). These data suggest that the aging-associated decline in neuronal function is dependent on odor concentration; consistent with previous studies showing relatively preserved behavioral detection of strong sensory stimuli with age (*Hummel et al., 2007*).

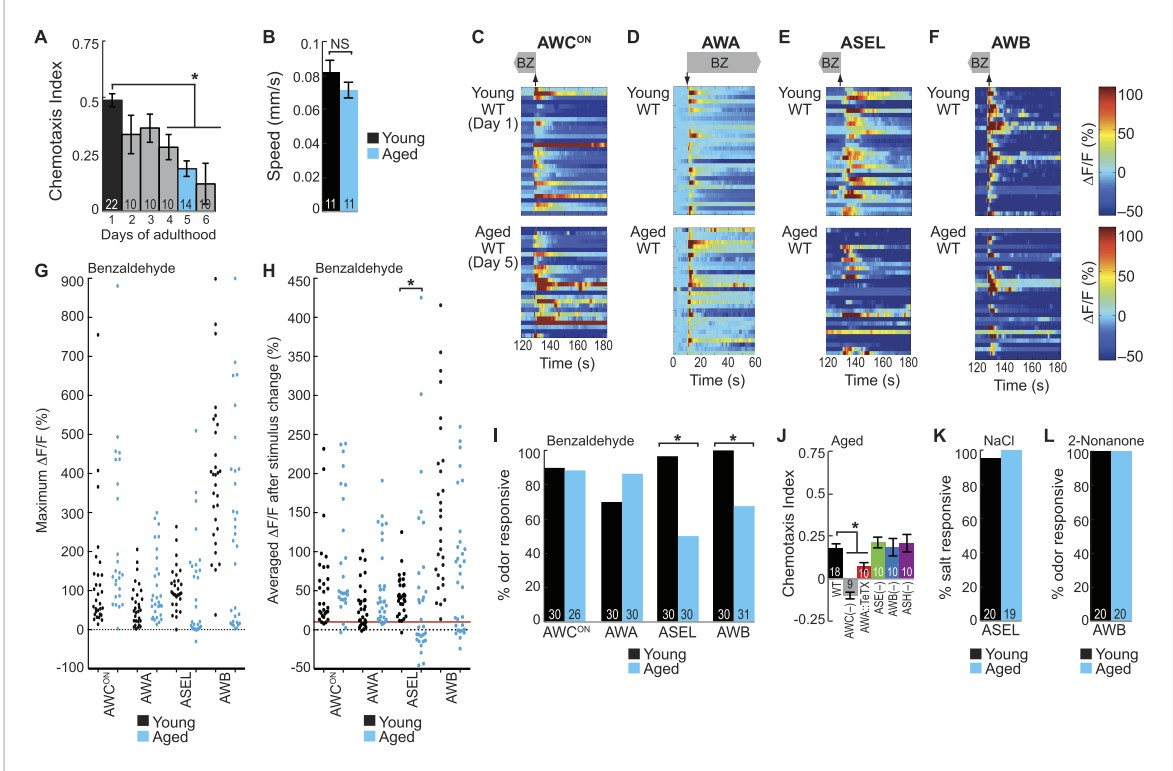

**Figure 5**. BZ-evoked secondary neuron activity and behavior specifically degrade with age. (**A**) Chemotaxis performance of wild-type worms from young adulthood (day 1) through early stage aging (day 6) towards a point source of medium BZ. (**B**) Speed of wild-type young (day 1) and aged (day 5) adult animals chemotaxing towards a point source of BZ odor. (**C–F**) Heat maps of ratio change in fluorescence to total fluorescence for wild-type young adult (day 1) and aged adult (day 5) sensory neuron responses to the addition (at t = 10 s) or removal (at t = 130 s) of a two-minute medium BZ stimulus (0.005% vol/vol), as indicated by shaded box and arrows. One row represents activity from one neuron. (**G**) Maximum ΔF/F for each individual young (black dots) or aged (blue dots) wild-type animal shown in **C–F**. (**H**) Averaged ΔF/F after odor addition (for AWA) or odor removal (for all other neurons) for each individual young (black dots) or aged (blue dots) wild-type animal shown in **C–F**. The red line represents a ΔF/F of 10%, the cutoff used to classify neurons as odor responsive or non-responsive. *p < 0.05, two-tailed *t*-test comparing young and aged responses; statistical analysis performed only on odor responsive subset of data. (**I**) Quantification of the percent of odor responsive neurons shown in **H**. (**J**) Aged (day 5) adult BZ chemotaxis performance of wild-type, AWC or AWB or ASH neuron-specific genetic ablation, AWA neuron-specific tetanus toxin expression worms or *che-1* mutants missing ASE neurons. (**K**, **L**) The percent of wild-type young (day 1) and aged (day 5) adult (**K**) ASEL neurons responsive to sodium chloride and (**L**) AWB neurons responsive to 2-nonanone odor. (**I**, **K**, **L**) Odor or salt responsive defined as having a ΔF/F to stimulus greater than 10%. Numbers on bars indicate number of neurons imaged. *p < 0.05, two-tailed Chi Square test. (**A**, **B**, **J**) Numbers on bars indicate number of assay plates and error bars indicate s.e.m. *p < 0.05, two-tailed *t*-test with Bonferroni correction, compared to young adults or wild-type as indicated. See *Figure 5—source data 1* for raw data.

The following source data and figure supplements are available for figure 5:

**Source data 1**. Age-related decay in odor responses and chemotaxis behavior data.
**Source data 2**. Primary and secondary neuron activity in young and aged animal data.
**Source data 3**. Correlated behavior and functional imaging in aged animal data.
**Source data 4**. Dose-dependent odor response data.
**Source data 5**. Salt and 2-nonanone responses in young and aged animal data.
**Source data 6**. Longevity mutant odor response data.

**Figure supplement 1**. Quantification of BZ-evoked primary and secondary neuron activity in young and aged animals.

**Figure supplement 2**. Olfactory behavior in aged animals is correlated with reliability of odor-evoked neuronal activity.

*Figure 5. continued on next page*

*Figure 5. Continued*

**Figure supplement 3**. Dose-dependent odor-evoked calcium dynamics in young and aged adults.
**Figure supplement 4**. ASE and AWB primary responses to salt and 2-nonanone, respectively, remain reliable with aging.
**Figure supplement 5**. Long and short-lived mutants do not influence the aging-associated declines in neuronal function.

Next, we investigated whether aging impairs all or only selective functions of ASEL and AWB neurons. To test this, we analyzed responses to salt (sodium chloride) and the repulsive odorant 2-nonanone, which are directly transduced by ASEL (*Bargmann, 2006*; *Suzuki et al., 2008*) and AWB neurons (*Troemel et al., 1997*), respectively. We found that neuronal activity and behavior in response to these stimuli remained reliable and robust in aged animals (*Figure 5K,L*, *Figure 5—figure supplement 4A–D*). These data indicate that functionality of both ASEL and AWB neurons in aged animals is sensory context dependent. Specifically, their primary responses to salt (ASEL) and 2-nonanone (AWB) are preserved, while their function as secondary neurons in encoding attractive BZ stimuli is impaired during aging.

We have previously shown AWC sensory neurons act as secondary neurons in the salt sensory circuit and respond to salt stimuli in an ASE-dependent manner (*Leinwand and Chalasani, 2013*). Therefore, we tested whether AWC secondary responses salt were also degraded during aging. However, we found that AWC responses to salt were not reduced in aged animals (*Figure 5—figure supplement 4E,F*). These data suggest that these early aging-associated deficits are specific to the BZ circuit, leaving the salt circuit fully functional. Together, these results show that there is a sensory context dependent decline in ASEL and AWB responses to BZ with age, disrupting the combinatorial code for attractive olfactory information specifically.

## Long-lived mutants do not affect aging-associated neuronal activity and behavioral declines

*C. elegans* is short lifespan model and has proven to be useful in identifying conserved organismal-level longevity pathways, such as insulin and energy and stress sensing pathways (*Wolff and Dillin, 2006*). We hypothesized that long-lived mutants might alter the dynamics of the age-associated decline in the combinatorial neural code for odor. We tested several distinct pathways shown to mediate lifespan extension. Gain of function (*gf*) mutants in the energy sensing alpha subunit of the AMP-activated protein kinase (AMPK, *aak-2* in *C. elegans* [*Apfeld et al., 2004*]) have increased lifespan. Similarly, animals without a germline due to ablation (*Hsin and Kenyon, 1999*) or mutations in the notch signaling pathway (*glp-1* in *C. elegans* [*Berman and Kenyon, 2006*]) have increased lifespan. Furthermore, whole animal RNAi treatment to knockdown the Rab-like GTPase *rab-10* also extends lifespan (*Hansen et al., 2005*). We recorded ASEL and AWB secondary neuron responses to BZ in aged day 5 adults in wild-type, long-lived *aak-2 (gf)* and *glp-1* mutants and *rab-10* knockdown animals. Similar to wild-type, the ASEL and AWB responses to medium BZ in *aak-2 (gf)* and *glp-1* mutants and *rab-10* knockdown animals were unreliable in day 5 adults (*Figure 5—figure supplement 5*). We also tested whether the aging-associated declines in olfactory behavior were altered in long-lived mutants. We found that *glp-1* mutants displayed a similar aging-associated decline in attraction to BZ compared to wild-type animals (*Figure 5—figure supplement 5I*). These data show that signaling from the longevity-modulating germline, AMP kinase energy sensing and Rab GTPase pathways do not attenuate secondary neuronal activity and behavior declines.

We also tested whether mutations that shorten lifespan could influence the aging-associated declines in neuronal function. A whole animal knockdown of the stress-induced heat shock factor 1 (*hsf-1*) was shown to be short-lived (*Hsu et al., 2003*). We found that animals with *hsf-1* knocked down had similarly unreliable day 5 aged ASEL and AWB secondary neuron responses to BZ compared to wild-type (*Figure 5—figure-supplement 5*). Taken together, these data suggest that the aging-associated declines in olfactory neuronal functions are independent of many known longevity pathways (*glp-1*, *aak-2*, *rab-10* and *hsf-1*).

## Aging-associated secondary ASEL activity declines are rescued by increased neurotransmission from AWC neurons

Our results show that the ASEL and AWB secondary neurons have unreliable odor-evoked activity in aged animals. This suggests that the neurotransmission that recruits these neurons to the odor circuit may break down with age. In particular, impaired ASEL neuronal activity may indicate a breakdown in the peptidergic neurotransmission that recruits this neuron into the BZ circuit. In order to identify the mechanisms for this aging-associated decline, we manipulated the primary to secondary neurotransmission pathway. First, we hypothesized that aging might downregulate the levels of the peptide receptors on ASEL neurons, thus reducing signaling in aged ASEL neurons. We tested this hypothesis by overexpressing the DAF-2 insulin receptor specifically in the ASEL neurons (*Figure 6A*, left panel). However, we found no change in the reliability of these aged animals' odor-evoked ASEL activity compared to wild-type (*Figure 6B,C,F*, *Figure 6—figure supplement 1B*). This result suggests that receptor expression is not limiting in these aged animals. We confirmed that our ASEL-specific DAF-2 overexpression (OE) was functional by analyzing ASEL responses in young day 1 adults. We found that the ASEL BZ responses were significantly larger in young adult animals overexpressing DAF-2 in ASEL (*Figure 6—figure supplement 1A,B*) confirming the efficacy of the transgene. Taken together, these results suggest that DAF-2 receptors in ASEL are not reduced during the aging process and signaling via these receptors does not limit olfactory circuit activity in aged animals.

We then tested whether primary AWC sensory neurons synthesize less neuropeptide as the animal ages, causing a breakdown in signaling to recruit ASEL neurons. To test this, we over-expressed the insulin-like neuropeptide INS-1 in the AWC neurons (*Figure 6A*, right panel). This manipulation succeeded in improving the reliability of odor-evoked activity in aged ASEL neurons, suggesting that increased neuropeptide production, and consequently release, can rescue aging-associated deficits (*Figure 6D,F*, *Figure 6—figure supplement 1B*). We also tested whether INS-1 (OE) could rescue aging-induced behavioral decline. We found that overexpressing INS-1 did not have a significant effect on the behavior of aged (or young) adults to BZ (*Figure 6—figure supplement 1D*). Together, these results show that while INS-1 (OE) can rescue the age-induced neural activity deficits, this is not sufficient to rescue aging-induced behavioral deficits. We suggest that the temporal properties of neuropeptide signaling are likely to be complex and that the INS-1 (OE) might have predicted effects on short timescales (a few seconds), but variable effects on longer timescales (hours to days).

To confirm a role for increased AWC neurotransmission in recruiting ASEL neurons, we also generated an AWC-specific RNAi knockdown of Tomosyn (*tom-1* in *C. elegans* [*Gracheva et al., 2007*; *Leinwand and Chalasani, 2013*]), a syntaxin-interacting protein that normally acts as a brake on all neurotransmission, to increase neuropeptide and neurotransmitter release from AWC neurons (*Figure 6A*, right panel). This manipulation to increase release from AWC neurons resulted in significantly more reliable odor-evoked ASEL activity (*Figure 6E,F*). These manipulations did not significantly affect ASEL responses in young day 1 adults (*Figure 6—figure supplement 1A–C*), suggesting that increased neurotransmission from the primary olfactory neurons specifically rescues the aging-associated ASEL defects. We also tested whether increased neurotransmission from AWC could rescue the aging-associated decline in chemotaxis behavior. We found that in aged day 5 adults, AWC-specific *tom-1* knockdown animals showed a significant improvement over wild-type in their attraction to BZ (*Figure 6G*). Moreover, this improvement required the presence of functional ASE neurons (*che-1* mutants do not have functional ASE neurons [*Uchida et al., 2003*]) (*Figure 6G*). Taken together, these results show that experimental manipulations to increase neurotransmission from AWC neurons rescue aging-induced decline in ASEL secondary neuron activity and animal behavior.

## Increasing AWA neurotransmission rescues aging-induced declines in AWB secondary neurons

We have shown that AWA neurons release acetylcholine, which is required for AWB neuronal activity in the young adult odor circuit. We hypothesized that this process could be reduced during aging; therefore, we tested whether manipulations to increase neurotransmission from AWA neurons could rescue the decline in aged AWB neural activity. We over-expressed the vesicular acetylcholine transporter, UNC-17, specifically in AWA neurons (*Figure 7A*). OE of the vesicular acetylcholine transporter was previously shown to increase the quantity of acetylcholine packed into and released from synaptic vesicles (*Song et al., 1997*). We found that this manipulation significantly increased the

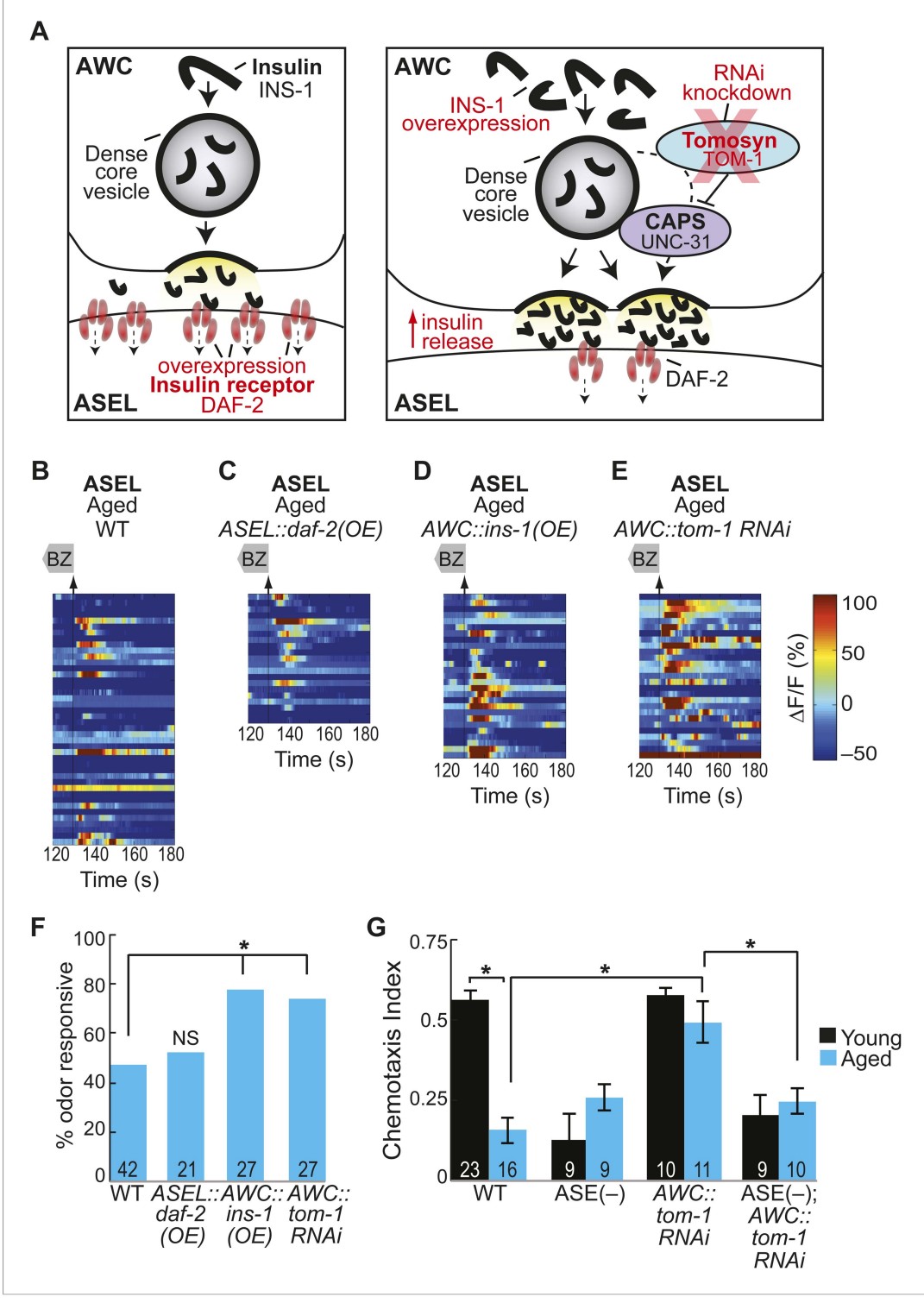

**Figure 6**. Increased neurotransmitter release from AWC neurons rescues aging-associated ASEL activity and behavioral deficits. (**A**) Schematic representation of genetic manipulations to overcome aging-associated decay of neurotransmission. (**B–E**) Heat maps of ratio change in fluorescence to total fluorescence for aged adult (day 5) ASEL sensory neuron responses to the removal (at t = 130 s) of a two-minute medium BZ stimulus (0.005% vol/vol) in (**B**) wild-type, (**C**) ASEL-specific *daf-2* overexpression (OE), (**D**) AWC-specific *ins-1* OE and (**E**) AWC-specific *tom-1* RNAi. (**F**) Quantification of the percent medium BZ responsive aged ASEL neurons in **B–E**. Odor responsive defined as having a ΔF/F to stimulus greater than 10%. Numbers on bars indicate number of neurons imaged.
*Figure 6. continued on next page*

*Figure 6. Continued*

*p < 0.05, two-tailed Chi Square test. (**G**) BZ chemotaxis in young and aged wild-type, *che-1* mutants lacking ASE neurons, AWC-specific *tom-1* RNAi and AWC-specific *tom-1* RNAi in the *che-1* background. *p < 0.05, two-tailed *t*-test with Bonferroni correction. See *Figure 6—source data 1* for raw data.

The following source data and figure supplement are available for figure 6:

**Source data 1**. Odor responses in AWC-released neurotransmitter manipulation animal data.
**Source data 2**. Additional odor responses in AWC-released neurotransmitter manipulation animal data.
**Figure supplement 1**. AWC-released neurotransmitters modify aging-associated neuronal activity and behavioral deficits.

reliability of aged AWB odor responses (*Figure 7B,C,E*, *Figure 7—figure supplement 1B*), further suggesting that increased signaling from the primary neurons can overcome aging-associated declines. We also confirmed a role for acetylcholine by using a pharmacological agent, arecoline. Arecoline is a cholinergic agonist known to act presynaptically to stimulate synaptic vesicle fusion (*Liu et al., 2013*) (*Figure 7A*). Acute arecoline treatment in aged animals significantly increased the probability of AWB odor responses (*Figure 7D,E*, *Figure 7—figure supplement 1B*), suggesting that a pharmacological approach to increase neurotransmission in aged animals can rejuvenate neuronal functions. Moreover, neither the UNC-17 OE nor acute arecoline had significant effects on AWB responses in day 1 adults (*Figure 7—figure supplement 1A–C*), confirming a specific role for increased neurotransmission in rescuing aged-associated AWB defects.

We also tested whether increased cholinergic transmission from the AWA neurons could rescue the aging-associated defects in behavioral attraction to BZ. We found that aged animals overexpressing the UNC-17 vesicular acetylcholine transporter in AWA neurons were significantly more attracted to BZ compared to aged wild-type animals (*Figure 7F*). Moreover, this increased attraction required the secondary AWB neurons (*Figure 7F*). These data confirm a role for AWA-AWB neurotransmission in rescuing aging-associated decline in BZ attraction. We note that while arecoline pharmacology rescued aged AWB neuronal activity, this treatment impaired BZ chemotaxis in both young and aged animals (*Figure 7—figure supplement 1D*). We suggest that the known effect of arecoline to increase spontaneous locomotion may be counterproductive to the directed locomotion required to chemotax up an odor gradient (*Glenn et al., 2004*; *Liu et al., 2013*).

## Aged animal olfactory behavior is correlated with lifespan

Finally, we investigated the consequences of individual variation in aged olfactory abilities at the whole animal level by testing whether the olfactory abilities we analyzed could be correlated with longevity. We performed chemotaxis assays and separated the animals into two populations that did or did not navigate up an attractive BZ gradient (*Figure 8A*). We then assayed the lifespan of these two populations of animals. Notably, we observed a significant extension (average of 16.2% in three separate trials, p < 0.001, Mantel–Cox test) in the lifespan of animals that successfully chemotaxed to the odor as aged adults, compared to animals that failed to do so (*Figure 8B*, *Figure 8—figure supplement 1*). However, we found no difference in the lifespan of animals that were sorted on the basis of their chemotaxis performance as young adults (*Figure 8C*, *Figure 8—figure supplement 1*). These results suggest that the olfactory abilities of aged, but not young, animals may be correlated with their overall health, leading to lifespan differences. In contrast, we found that sorting aged animals based on their attraction to salt did not result in any significant differences in lifespan (*Figure 8—figure supplements 1,2A*). These data show that the increase in lifespan is likely to be specific to BZ and not the salt associated neural circuit, consistent with the specific declines in BZ, not salt, evoked activity and behavior. Furthermore, these results indicate that the functionality of some, but not all, sensory neuronal circuits in early stage aged animals may predict animals' longevity. These data are also consistent with cell ablation experiments where loss of some chemosensory neurons affects *C. elegans* lifespan, while loss of other chemosensory

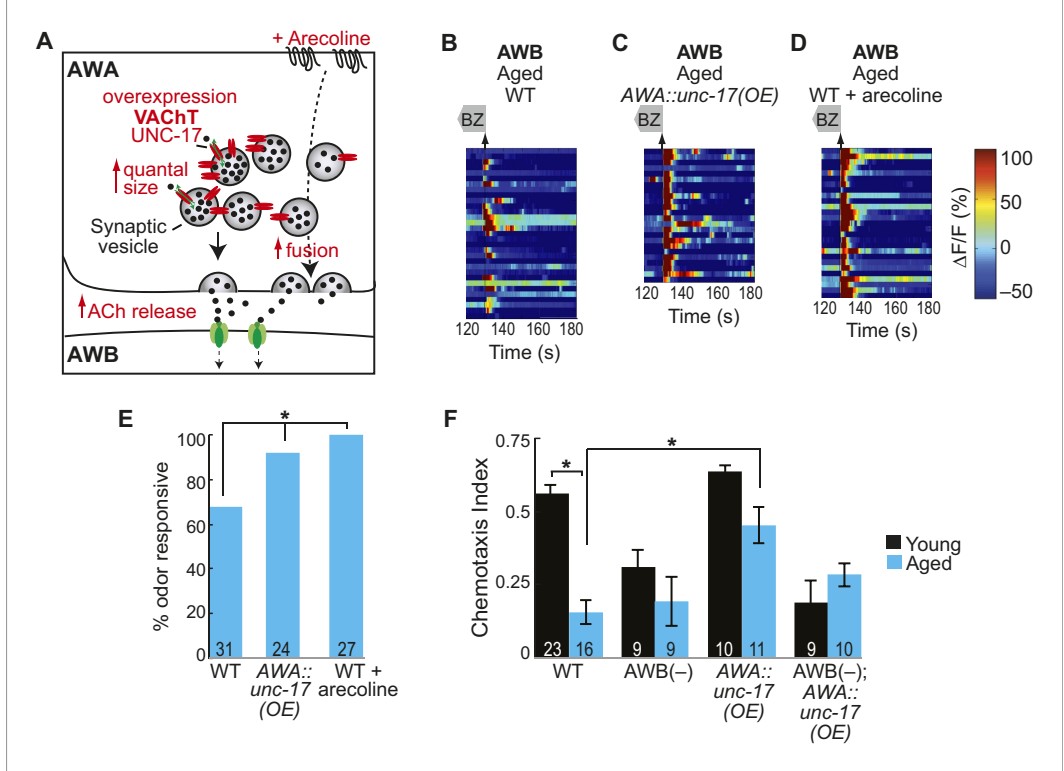

**Figure 7**. Increased release from AWA primary neurons rescues aging-associated AWB activity and behavioral deficits. (**A**) Schematic representation of genetic and pharmacologic manipulations to overcome aging-associated decay of neurotransmission. (**B–D**) Heat maps of ratio change in fluorescence to total fluorescence for aged adult (day 5) AWB sensory neuron responses to the removal (at t = 130 s) of a two-minute medium BZ stimulus (0.005% vol/vol) in (**B**) wild-type, (**C**) AWA-specific *unc-17* OE, and (**D**) animals treated acutely with the cholinergic agonist arecoline. (**E**) Quantification of the percent BZ responsive aged AWB neurons in **B–D**. Odor responsive defined as having a ΔF/F to stimulus greater than 10%. *p < 0.05, two-tailed Chi Square test. (**F**) Medium BZ chemotaxis in young and aged wild-type, AWB neuron ablated, AWA-specific *unc-17* OE and AWB ablated in the AWA-specific *unc-17* OE background. *p < 0.05, two-tailed *t*-test with Bonferroni correction. See *Figure 7—source data 1* for raw data.

The following source data and figure supplement are available for figure 7:

**Source data 1**. Odor responses in AWA-released neurotransmitter manipulation animal data.

**Source data 2**. Additional odor responses in AWA-released neurotransmitter manipulation animal data.

**Figure supplement 1**. AWA neurotransmission modifies aging-associated neuronal activity and behavioral deficits.

neurons has no effect (*Alcedo and Kenyon, 2004*). Together, these results suggest that the olfactory prowess of aged animals is indicative of whole animal physiology, health and lifespan.

We next investigated the mechanisms linking aged olfactory abilities and longevity. We tested whether more reliable olfactory circuit functioning resulting from increased neurotransmission from the primary AWA or AWC[ON] neurons affected animal lifespan. We found that animals overexpressing the UNC-17 vesicular acetylcholine transporter in AWA neurons lived an average of 26.6% longer than their wild-type counterparts (over three separate trials, p < 0.001, Mantel–Cox test) (*Figure 8D*, *Figure 8—figure supplement 3*). Moreover, increasing neurotransmission from AWC either by overexpressing the INS-1 peptide or by knocking down *tom-1*, the *C. elegans* homolog of Tomosyn, resulted in a trend towards a small extension in lifespan (*Figure 8E*, *Figure 8—figure supplement 3*). These data suggest that both classical neurotransmission (from AWA neurons) and neuropeptide signaling (from AWC neurons), which are key components of

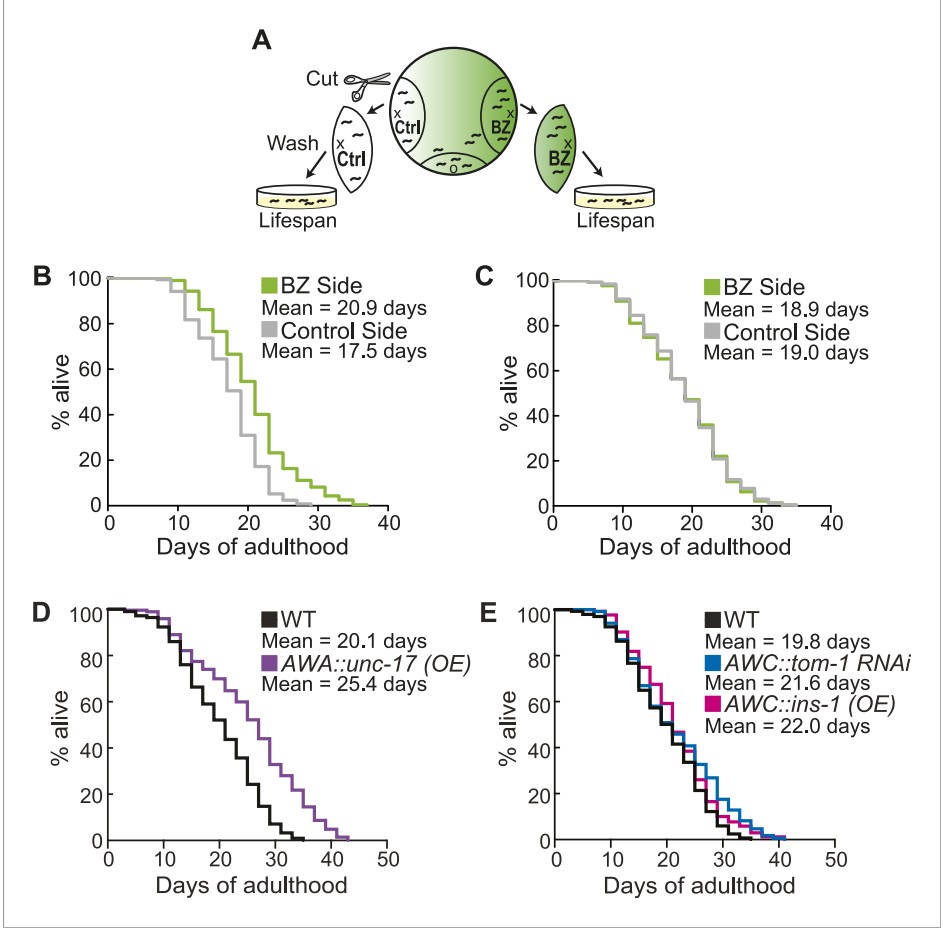

**Figure 8**. Aged animal olfactory abilities and neurotransmission from primary neurons are correlated with lifespan.
(**A**) Schematic of animals from a chemotaxis assay washed and sorted into two populations based on successful or
failed navigation up the BZ odor gradient, for lifespan analysis. (**B**) Animals that chemotaxed to the BZ odor side of
the chemotaxis plate as aged adults (day 5) have a 16.2% average extension in their lifespan compared to animals
from the opposite, control (Ctrl) side (p < 0.01 by Mantel–Cox test, see *Figure 8—figure supplement 1* and
*Figure 8—source data 1* for quantification). (**C**) Animals sorted by their young adult chemotaxis do not have
significantly different lifespans (see *Figure 8—figure supplement 1* and *Figure 8—source data 1*). (**D**) AWA-
neuron specific *unc-17* OE transgenic animals have a 26.6% average extension in lifespan compared to wild-type
animals (p < 0.0001 by Mantel–Cox test, see *Figure 8—figure supplement 3* and *Figure 8—source data 1* for
quantification). (**E**) Survival of wild-type, AWC-neuron specific *tom-1* RNAi, and AWC-specific *ins-1* OE transgenic
animals (see *Figure 8—figure supplement 3* and *Figure 8—source data 1* for quantification). (**B–E**) Mean survival
is reported in days of adulthood. BZ, benzaldehyde; OE, overexpression.
The following source data and figure supplements are available for figure 8:

**Source data 1**. Lifespan of animals sorted by their chemotaxis performance and lifespan of neurotransmitter
manipulation transgenic animal data.

**Source data 2**. Additional lifespan of neurotransmitter manipulation transgenic animal data.

**Figure supplement 1**. Sorting animals based on their performance on odor chemotaxis affects lifespan.

**Figure supplement 2**. Sorting animals based on their performance on salt chemotaxis and silencing primary
neurons modifies lifespan.

**Figure supplement 3**. Manipulating neurotransmission from primary olfactory neurons modifies lifespan.

the combinatorial code for BZ, may have a longevity promoting effect. These data are in contrast with previously published results showing that animals with AWC or AWA neurons ablated live longer (*Alcedo and Kenyon, 2004*); therefore, we probed the role of these neurons in lifespan further. We analyzed the lifespan of animals in which AWA or AWC neurons were silenced rather than ablated. Animals expressing tetanus toxin in either AWA or AWC neurons to block their neurotransmission lived significantly longer than wild-type (average of 36.0% and 27.6% longer, respectively, over two independent trials, p < 0.01, Mantel–Cox test, *Figure 8—figure supplement 2B,3*). Together, these results suggest that a balance in neurotransmission from the primary olfactory neurons is crucial to an animal's longevity; both higher than normal and lower than normal levels of neurotransmission extend lifespan. We suggest that signaling from these primary olfactory neurons is integrated by the downstream circuitry to mediate effects on the animal's lifespan.

## Discussion

Our results define a novel neural circuit mechanism for encoding sensory information to drive behavior and demonstrate age-related functional declines in this circuit. These data provide the first indication that *C. elegans* employ a combinatorial olfactory coding strategy as in flies and mice, suggesting that this strategy is essential for behavioral plasticity (*Wang et al., 2003*; *Oka et al., 2006*). Moreover, we suggest that primary olfactory neurons directly detect odors and use neurotransmission to recruit additional secondary neurons. However, activity in the secondary neurons declines with aging, leading to behavioral deficits.

We propose that the combination of primary and secondary neurons may be a common motif in sensory neural circuits from worms to mammals. A distinct, but similarly distributed neural circuit (which does not include the ASE neurons) encodes a different attractive odorant, isoamyl alcohol (data not shown, [*Yoshida et al., 2012*]). Furthermore, we have previously shown that the *C. elegans* salt neural circuit is composed of a primary salt sensory neuron, ASEL, which releases INS-6 insulin neuropeptides to recruit a secondary sensory neuron, AWC$^{ON}$, into the circuit in particular sensory contexts (*Leinwand and Chalasani, 2013*). This combined primary and secondary neuron coding strategy is likely to increase the signal-to-noise ratios, thus preventing failures in encoding sensory information. Combinatorial coding of this sort may also be broadly useful for distinguishing different concentrations of the same stimulus, as they will be encoded by overlapping but distinct subsets of neurons. This approach may also enhance the ability of young adults to successfully find food, perhaps to enhance reproductive success, while the aging-associated declines occur in post-reproductive animals that may have reduced nutritional demands. Furthermore, the insulin peptidergic and cholinergic signaling from primary to secondary olfactory neurons could add salience to volatile food signals in a complex, multisensory environment. Previous studies have shown that insulin (*Lacroix et al., 2008*) and cholinergic receptors (*Ogura et al., 2011*) are expressed in mammalian olfactory processing centers, suggesting that these signaling pathways might also be used to encode odors in mammals. Detailed analyses of the architecture of sensory circuits, including the neurotransmission between sensory neurons, in other species are needed to determine whether the circuit motif described here is broadly conserved.

We find that primary sensory transduction remains robust as animals age. However, the combinatorial code for attractive volatile cues degrades with age because the activity of cells functioning as secondary neurons decays with age. Our results show that aging-induced decline in neuronal function is dependent on the interplay between sensory context and neuronal identity. For example, we find that primary ASEL responses to salt and AWB responses to 2-nonanone are preserved while their secondary responses to BZ are reduced in aged animals. This is in contrast with studies showing an early stage age-induced decline in the primary ASH neuron responses to hyperosmotic stimuli (*Chokshi et al., 2010*). Taken together, these results indicate that aging differentially affects sensory circuits, perhaps reflecting differences in physiological demand and the importance of diverse sensory contexts as the animal ages. Furthermore, these aging-associated sensory declines occur independently of many known longevity pathways. Insulin signaling has been shown to promote longevity in a number of model systems including *C. elegans*, *Drosophila melanogaster* and *Mus musculus* (*Broughton and Partridge, 2009*; *Kenyon, 2010*). We find that insulin signaling is required for the primary AWC$^{ON}$ to secondary ASEL neurotransmission and so are unable to separate its longevity promoting effect from its role in encoding sensory information. We speculate that the insulin signaling pathway might affect both the quality of an animal's life by encoding odors based on sensory context and also its lifespan.

Our experiments show that experimental manipulations targeting neurotransmission pathways improve the aging-associated neuronal activity and olfactory behavioral declines. Several different mechanisms could underlie the impairments observed in aged animals and overcome by our manipulations. A decline in peptidergic and cholinergic gene expression with age could contribute; however, quantitative RT-PCR experiments suggest that there is no aging-associated reduction in the expression of these genes at the whole animal level [data not shown and (*Jin et al., 2011*)]. Changes in gene expression specifically in the primary olfactory neurons cannot be ruled out. Nevertheless, we speculate that the early aging-associated sensory impairments are driven at least in part by reduced neurotransmitter release from primary neurons, a mechanism likely applicable across species. We find that both increasing neurotransmitter production and release capacity rescue the aging-associated deficits. Therefore, it is likely that aging affects multiple steps in the neurotransmitter release pathway, emphasizing the key role played by this machinery in regulating animal behavior and physiology. These results are consistent with reports of reduced synapse number in the aged mammalian olfactory bulb, which should disrupt olfactory circuits (*Richard et al., 2010*). We speculate that these differences in synaptic transmission also explain some of the inter-individual variability in aging phenotypes (*Pinto et al., 2014*; *Vijg, 2014*). Subsequently, these circuit-level changes could produce hyposmia or anosmia, which may be among the earliest predictors of lifespan and mortality across species (*Toth et al., 2012*; *Liu et al., 2013*; *Pinto et al., 2014*). More generally, we suggest that alterations in transmitter release, which disrupt neuronal communication throughout the brain (*Dickstein et al., 2007*) are likely to underlie variability in individual animal behavior and age-related cognitive and behavioral decline.

## Materials and methods

*C. elegans* strains were grown and maintained under standard conditions (*Brenner, 1974*). A complete listing of all strains used in this study and their genotypes is located in *Supplementary file 1*.

### Molecular biology and transgenesis

cDNA corresponding to the entire coding sequences of *unc-31* (isoform a), *daf-2* (isoform a), *age-1* (isoform a), *tom-1* (isoform a), and the *ins-1* genomic region were amplified by PCR and expressed under cell-selective promoters. *unc-17* cDNA was synthesized (GenScript) and expressed under a cell-selective promoter. For *cha-1* and *cho-1* knockdown experiments, 1 kb fragments corresponding to exons 3–7 and the 3′ end of the gene, respectively, in the sense and antisense orientation were synthesized (GenScript). Neuron-selective RNAi transgenes were created as previously described by co-injection of equal concentrations of sense and antisense oriented gene fragments driven by cell-specific promoters (*Esposito et al., 2007*). Cell-specific expression was achieved using the following promoters: *ceh-36deletion* or *odr-3* for both AWC, *str-2* for AWC[ON], *srsx-3* for AWC[OFF], *gpa-4* for AWA and ASI, *gpa-4deletion* for AWA, *gcy-7* for ASEL, *gcy-5* for ASER, *str-1* for AWB, *sre-1* for ADL, *srh-142* for ADF, *gcy-8* for AFD, *ops-1* for ASG, *sra-6* for ASH, *trx-1* for ASJ and *sra-9* for ASK. For all experiments, a splice leader (SL2) fused to a *mCherry* or *gfp* transgene was used to confirm cell-specific expression of the gene of interest.

Germline transformations were performed by microinjection of plasmids (*Mello and Fire, 1995*) at concentrations between 25 and 200 ng/μl with 10 ng/μl of *unc-122::rfp*, *unc-122::gfp* or *elt-2::gfp* as co-injection markers. For rescue and OE experiments, DNA was injected into mutant or wild-type *C. elegans* carrying GCaMP arrays.

### Calcium imaging

Transgenic worms expressing GCaMP calcium indicators under a cell-selective promoter were grown to day 1 or day 5 of adulthood and trapped in a custom designed PDMS microfluidic device and exposed to odor stimuli (*Chalasani et al., 2007*; *Chronis et al., 2007*). For aging experiments, a new PDMS device with larger channels was designed to trap and stimulate day 5 adult worms (*Chokshi et al., 2010*). Older, day 6 adult worms exhibit much larger variation in whole animal size than day 5 adults (see *Figure 5—figure supplement 1A*) and could not be trapped consistently without introducing bias into the experiment. For aging experiments, animals were transferred to new OP50 bacteria plates every other day to track the aging animals and to avoid contamination by their progeny. Additionally, for whole animal RNAi experiments to knockdown *rab-10* and *hsf-1*, animals

were fed either control (Ctrl) empty pL4440, *rab-10 RNAi* or *hsf-1 RNAi* expressing bacteria beginning at day 1 of adulthood as previously described (*Hansen et al., 2005*).

Fluorescence from the neuronal cell body was captured using a Zeiss inverted compound microscope for 3 min. We first captured 10 s of baseline activity (t = 0–10 s) in chemotaxis assay buffer (5 mM $K_3PO_4$ (pH 6), 1 mM $CaCl_2$, 1 mM $MgSO_4$, and 50 mM NaCl), then 2 min (t = 10–130 s) of exposure to an odor (or salt) stimulus dissolved in chemotaxis buffer, and lastly 50 s (t = 130–180 s) of buffer only. BZ refers to a 0.005% vol/vol dilution in chemotaxis assay buffer, except where low BZ (0.0001% vol/vol) or high BZ (0.1% vol/vol) is specifically mentioned. Additionally, a 0.1% vol/vol dilution of 2-nonanone and 50 mM sodium chloride stimulus were used as indicated. For arecoline experiments, worms were pre-treated with 0.15 mM arecoline in chemotaxis buffer for approximately 20 min and immediately imaged in the presence of the drug. Laser ablations of the paired AWC, AWA, AWB or ASE sensory neurons, along with mock ablations, were performed as previously described (*Bargmann and Avery, 1995*) in transgenic animals expressing GCaMP. In all experiments, a single neuron was imaged in each animal, and each animal was imaged only once. Wild-type Ctrls, mutants, and transgenic or drug treated strains for each figure were imaged in alternation, in the same session.

We used Metamorph and an EMCCD camera (Photometrics) to capture images at a rate of 10 frames per second. A MATLAB script was used to analyze the average fluorescence for the cell body region of interest and to plot the percent change in fluorescence for the region of interest relative to $F_0$, as previously described (*Chalasani et al., 2007*). Specifically, data was plotted and statistical analysis was performed as follows: (1) for line graphs of ΔF/F over time (*Figures 1–4* and corresponding figure supplements), the average fluorescence in a 8 s window (t = 1–9 s) was set as $F_0$. Average and standard error at each time point were generated and plotted using MATLAB scripts, as previously described (*Leinwand and Chalasani, 2013*). (2) For heat maps (*Figures 5–7* and corresponding figure supplements), the average fluorescence in a 8 s window (t = 1–9 s) was set as $F_0$.

To quantify calcium responses, $F_0$ was consistently set to the average fluorescence signal from 1 s to 9 s prior to the relevant change (addition or removal) of stimulus. For statistical analysis, the average fluorescence and standard error were calculated for each animal over a short period corresponding to the duration of a response. Specifically, to analyze on responses to the addition of stimulus, the average fluorescence and standard error were calculated in the 10 s period following the addition of odor or salt (t = 10–20 s). For AWA neurons, the response duration was very brief; therefore, a 4 s time period was used instead (t = 10–14 s) so that small, fast responses could be appropriately quantified. To analyze off responses to the removal of stimulus, the average fluorescence and standard error were calculated in the period following the removal of odor (t = 130–140 s for all cells except ASE, and t = 130–145 for the slower, longer duration ASE responses). Traces in which an averaged ΔF/F of greater than 600% was recorded were excluded as they are likely to be artifacts of the neurons moving out of the focal plane and these usually account for less than 1% of the traces collected. To determine whether there was an odor-evoked increase or suppression of the calcium signal (see *Figure 1C*), the average fluorescence in these time windows in buffer only trials was compared (by a two-tailed unpaired *t*-test) to the average fluorescence in odor stimulation trials, for each neuron. The maximum ΔF/F in these time periods following odor addition or removal and the time to reach this maximum ΔF/F (from the stimulus change, in seconds) were also quantified (see *Figure 5G*, *Figure 1—figure supplement 1A,B* and *Figure 5—figure supplement 1B*). More specifically:

(1) For bar graphs of averaged ΔF/F after odor addition or removal (*Figures 2–4*, *Figure 3—figure supplement 1* and *Figure 4—figure supplement 1*): (a) $F_0$ was set to the average fluorescence from 1–9 s for quantification of AWA neuron responses to the addition of BZ stimulus and (b) $F_0$ was set to the average fluorescence from 121–129 s for quantification of AWC, ASE and AWB responses to the removal of BZ or 2-nonanone. Two-tailed unpaired *t*-tests were used to compare the responses of different genotypes or cell ablation conditions, and the Bonferroni correction was used to adjust for multiple comparisons.

(2) For scatter plots of maximum ΔF/F (*Figure 1—figure supplement 1A* and *Figure 5G*) and scatter plots of averaged ΔF/F after stimulus change (*Figure 5H*, *Figure 6—figure supplement 1B* and *Figure 7—figure supplement 1B*): (a) for AWA neurons' response to the addition of odor stimulus $F_0$ was set to the average fluorescence from 1–9 s and (b) for $AWC^{ON}$, ASEL and AWB responses to odor stimulus removal $F_0$ was set to the average fluorescence from 121–129 s. For the subset of odor-responsive neurons (exceeding the 10% ΔF/F cut-off), the averaged ΔF/F after the

stimulus change and the time to the maximum $\Delta F/F$ were also analyzed using two-tailed unpaired $t$-tests to compare different ages or genotypes (*Figure 5H*, *Figure 5—figure supplement 1B*, *Figure 6—figure supplement 1B* and *Figure 7—figure supplement 1B*). Furthermore, considering only the odor responsive neurons, no significant differences were observed in the magnitude of the odor-evoked suppression of young and aged animals (comparing the average fluorescence in ten second windows tiling the period of odor stimulation, by two-tailed $t$-test), indicating that our subsequent analyzes of the odor removal time period are not biased by the choice of the $F_0$.

(3) For bar graph quantifications of the % odor or salt responsive neurons in the aging experiments (*Figure 5I,K,L, 6F, and 7E* and the corresponding figure supplements): (a) $F_0$ was set to the average fluorescence from 1–9 s for quantification of the percent of AWA and ASH neurons responsive to the addition of BZ stimulus and for the percent of ASEL and AWC neurons responsive to the addition of NaCl salt stimulus. (b) $F_0$ was set to the average fluorescence from 121–129 s for AWC[ON], ASEL and AWB responses to BZ or 2-nonanone odor stimulus removal. The percent of odor responsive neurons was calculated by determining the proportion of cells displaying an average fluorescence ($\Delta F/F$) greater than 10% after odor addition (for AWA and ASH) or odor removal (all other neurons). 10% $\Delta F/F$ was used as the cut-off for odor responsiveness because, for all neurons imaged, changing buffer around the nose of the animal elicited a response smaller than this cut-off. Similarly, neurons displaying an average fluorescence ($\Delta F/F$) greater than 10% after salt addition were considered to be salt responsive. A two-tailed Chi–Square test was used to compare the percent of odor or salt responsive neurons in different conditions.

## Chemotaxis assays

Odor chemotaxis assays were performed as previously described (*Ward, 1973*). For aging assays, worms were synchronized by hatch offs in which 8 young adult worms were given 150 min to lay eggs on a large plate before being picked off. These eggs were grown at 20° until the appropriate day of adulthood, except for *glp-1* mutants, which were raised at the restrictive temperature, 25°. Aging animals were transferred to new bacteria plates every other day to track the aging animals and to avoid contamination by their progeny. Chemotaxis assays were performed on 2% agar plates (10 cm diameter) containing 5 mM potassium phosphate (pH 6), 1 mM $CaCl_2$ and 1 mM $MgSO_4$. Animals were washed once in M9 and three times in chemotaxis buffer (5 mM $K_3PO_4$ (pH 6), 1 mM $CaCl_2$ and 1 mM $MgSO_4$). For arecoline chemotaxis experiments, 0.15 mM arecoline was added to the M9 and chemotaxis buffer washes, yielding a 16–20 min drug treatment immediately prior to the behavioral experiment. Odor concentration gradients were established by spotting diluted BZ (0.2% vol/vol, in ethanol) near the edge of the plate, with a Ctrl 1 μl of ethanol spotted at the opposite end of the plate. Where noted, 1 μl of neat BZ was used for high concentration point source assays. For 2-nonanone experiments, a 50% vol/vol dilution of 2-nonanone in ethanol was used. For salt chemotaxis experiments, salt gradients were established by placing a Ctrl or a high salt (500 mM NaCl) agar plug on the assay plate and allowing 16–20 hr for the salt to diffuse and form a gradient (*Leinwand and Chalasani, 2013*). 1 μl of sodium azide was added to the odor (or salt) and the Ctrl spots to anesthetize animals reaching the end points. Washed worms were placed on the plate and allowed to move freely for one hour. The chemotaxis index was computed as the number of worms in the region near the odor (or salt) minus the worms in the region near the Ctrl divided by the total number of worms that moved beyond the origin. Nine or more assays were performed, over at least three different days. Two-tailed unpaired $t$-tests were used to compare the responses of different genotypes or ages, and the Bonferroni correction was used to adjust for multiple comparisons.

## Correlated chemotaxis and imaging experiments

Transgenic worms bearing GCaMP arrays, synchronized by a hatch off as described above, were grown until day 5 of adulthood at 20°. Aging animals were transferred to new bacteria plates every other day to track the aging animals and to avoid contamination by their progeny. Animals were tested in a (0.2% vol/vol) BZ odor chemotaxis assay as above, with two modifications. First, no sodium azide was used to paralyze the animals. Second, animals were given only 30 min to move freely on the chemotaxis plate. The chemotaxis assay plate was then cut into three regions corresponding to the BZ odor side, the middle, and the ethanol Ctrl region immediately after 30 min and worms were washed off each section separately and allowed to recover on OP50

bacteria plates for at least 90 min. Worms from the odor and the Ctrl sections of the chemotaxis assay were imaged in alternation as described above.

## Lifespan assays

Worms, synchronized by a hatch off as described above, were grown until day 1 or 5 of adulthood at 20°. To sort animals on the basis of their chemotaxis performance, wild-type animals were tested in a (0.2% vol/vol) BZ odor or (500 mM NaCl) salt chemotaxis assay as above, but without sodium azide and with only 30 min for the animals to move freely on the chemotaxis plate. The chemotaxis assay plate was then cut into a BZ odor (or salt), middle, and Ctrl region and worms were washed off each section separately. 100 adults from the odor (or salt) region or the Ctrl region were transferred onto 10 small OP50 plates (10 adults per plate) and grown at 20°. For experiments with transgenic animals, day 1 animals bearing the appropriate transgene were picked from the hatch off plate directly onto 10 small OP50 plates (10 adults per plate) and grown at 20°. In all experiments, aging animals were transferred to new bacteria plates every other day to track the aging animals and to avoid contamination by their progeny. Survival was analyzed every other day and worms were scored alive or dead based on their response to a gentle head touch (or lack thereof) as previously described (*Kenyon et al., 1993*). Worms were censored if they bagged, exploded or desiccated on the side of the plate. The chemotaxis assay followed by lifespan analysis or lifespan assays with transgenic animals were repeated two or three times per condition as indicated, beginning on separate days. The percent change in mean survival was calculated as the mean survival of animals from the odor side minus the mean survival of animals from the Ctrl side divided by the mean odor side survival or the mean transgenic animal survival minus the mean wild-type survival divided by the mean wild-type survival. Statistical analysis of lifespan was performed by the Mantel–Cox Log–Rank test, using GraphPad Prism and OASIS (*Yang et al., 2011*).

## Speed analysis

Chemotaxis assays to BZ were set up as described above, but with modifications to enable automated analysis of animal speed. 200 mM $Cu(II)SO_4$-soaked filter paper was placed on a standard chemotaxis assay plate to contain the worms in a reduced chemotaxis arena (1.25 by 1.25 inch square). 1 μl of BZ (0.2% vol/vol dilution in ethanol) and a Ctrl 1 μl of ethanol were spotted at opposite corners of the square arena, without any paralytic. After washing, only 5 worms were placed on the chemotaxis plate; this number minimized collisions and enabled more accurate tracking. The movement of the animals was tracked over 60 min using a Pixelink camera and speed was analyzed using previously published MATLAB scripts to track the centroid of the animal (*Ramot et al., 2008*). The results from eleven chemotaxis plates were averaged for each age. NS indicates p > 0.05, two-tailed *t*-test.

## Aged worm measurements

Day 5 and day 6 adult worms from hatch offs performed on three separate days were immobilized with tetramisole and imaged on 2% agarose pads. Images were captured on a Zeiss Observer D1 microscope using a 10× objective with DIC. The perimeters of 55 worms were measured using MetaMorph software.

## Acknowledgements

We thank the *Caenorhabditis* Genetics Center, the National Brain Research Project (Japan), C Bargmann, P Sengupta, Y Iino, I Maruyama, D Kim, P Sternberg and A Zaslaver for worm strains. We thank M Ailion for *unc-31* cDNA, P Sengupta for neuron specific promoters and L Stowers, C Stevens, J Wang, Y Jin, M Hansen, L Hale, K Quach and C Yeh for helpful discussions and comments on the manuscript. We are also grateful to A Tong, Z Liu and other members of the Chalasani laboratory for their help and advice. Grants from the Rita Allen Foundation, the WM Keck Foundation, the NIH (R01MH096881-03) (SHC) and a National Science Foundation Graduate Research Fellowship and Achievement Rewards for College Scientists Scholarship (SGL) funded this work. The content is solely the responsibility of the authors and does not necessarily represent the official views of the National Institutes of Health and the National Science Foundation.

## Additional information

### Funding

| Funder | Grant reference | Author |
|---|---|---|
| National Institutes of Health | R01MH096881-03 | Sreekanth H Chalasani |
| Rita Allen Foundation | | Sreekanth H Chalasani |
| W.M. Keck Foundation | | Sreekanth H Chalasani |
| National Science Foundation | | Sarah G Leinwand |

The funders had no role in study design, data collection and interpretation, or the decision to submit the work for publication.

### Author contributions

SGL, Conception and design, Acquisition of data, Analysis and interpretation of data, Drafting or revising the article; CJY, Acquisition of data, Analysis and interpretation of data; DB, NC, Designed and developed a novel microfluidic chip for trapping aged animals, Contributed unpublished essential data or reagents; JS, Performed all the neuron ablations in the manuscript, Drafting or revising the article, Contributed unpublished essential data or reagents; SHC, Conception and design, Analysis and interpretation of data, Drafting or revising the article

### Ethics

Animal experimentation: This study was performed in strict accordance with the recommendations in the Guide for the Care and Use of Laboratory Animals of the National Institutes of Health.

## Additional files

### Supplementary file

• Supplementary file 1. *C. elegans* strain list.

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
