## [Decision Letter]

Thank you for resubmitting your work entitled “Circuit mechanisms encoding odors and driving aging-associated behavioral declines in *Caenorhabditis elegans*” for further consideration at *eLife*. Your article has been favorably evaluated by a Senior Editor, a Reviewing Editor, and two reviewers. The manuscript has been improved but there are some remaining issues that need to be addressed before acceptance, as outlined below. As you can tell these issues require some significant re-writing, but no further experimentation.

1) AWB shows a nice odor-off response even in AWA-ablated animals. The source of this sensory stimulus is not known, one possibility being redundant inputs from AWA and AWC. Small but significant AWB response is also observed in the *unc-13* mutant, *unc-17* mutant and other manipulations. Given the status of the analysis and the characteristics of AWB as olfactory neurons, direct sensory input to AWB is not excluded and is in fact likely. Please discuss this issue in the text and put an arrow (maybe dotted arrow) from “benzaldehyde” to “AWB” in Figures 2 and 4, unless you identify neurons that account for 100% of the odor-off response of AWB.

2) Also, AWA's role is enigmatic. Upon odor removal, at which time AWB is activated, AWA does not respond at all, or show only a small calcium increase. On the contrary, upon odor addition, AWA is largely activated but AWB is not activated, or a bit inhibited. Therefore a simple excitatory or inhibitory synapse from AWA to AWB does not explain the behavior of AWB neurons. This simple consideration leads to the question whether AWBs dependence on AWA is based on fast synaptic transmission. You show that the *unc-17* mutant shows reduced response of AWB and this defect is rescued by *unc-17* expression in AWA. *cho-1* RNAi shows similar results. However, this does not necessarily mean AWB is activated upon odor removal by cholinergic neurotransmission from AWA. Equally possible explanations – which you should discuss in the text – are the modulatory effects of AWA-released acetylcholine (as muscarinic receptors function in other systems), by which AWB becomes more responsive to direct odor sensation. In the Abstract, you state: “Primary neurons, AWC^ON^ and AWA, directly detect the food odor benzaldehyde and release insulin-like peptides and acetylcholine, respectively, to activate secondary neurons, ASEL and AWB”. This clearly implicates that insulin and acetylcholine are released upon odor detection to activate ASEL and AWB. This is not supported because there is no release timing data presented for neither insulin nor acetylcholine (these data are generally difficult to obtain in *C. elegans*). You need to carefully rephrase these statements to something like “insulin-like peptide and acetylcholine released from AWC and AWA respectively are required for activity of secondary neurons”.

3) The quantification data in Figure 5—figure supplement 1 are persuasive in showing that only ASEL and AWB decline by age. However, you need to accurately describe the quantification method. In the Methods, you state: “To analyze off responses to the removal of stimulus, F0 was set to the time just prior to odor removal (t=121-129s)”. This is probably not what you did for the heat maps in Figure 5, Figure 6, Figure 7, which show only the 120s-180s time points, and the dot plots in Figure 5—figure supplement 1. Instead, F0 seems to come from the average of t=1-9s for these analyses.

4) In the Methods, you state: “the percent of odor responsive neurons was calculated by determining the proportion of cells displaying an average fluorescence (ΔF/F) greater than 10 percent after odor addition (for AWA) or odor removal (all other neurons)”. For Figure 5, Figure 6, Figure 7 and other similar figures, which time points did you use for calculating F0 (or F), 1-9s or 121-129s?

The definition of F0, of course, affects the interpretation of the results. All time series traces of ASEL and AWB calcium levels, like those in Figure 2 show downward trends during odor exposure. If F0 is defined for 1-9s, all data quoted above might mean that aged ASEL and AWB are more sensitive to suppression by sustained odor exposure. This can be appreciated by quantifying the decline in calcium levels during 2-minutes odor exposure in young and aged animals, but this result is not presented and it should be.

It is fine either way, as far as there is a change in “secondary neuron” responses between young and aged animals, but the results must be accurately presented so that readers are not lead to misunderstandings.

5) Please indicate whether you tested whether *ins-1* mutants are defective in chemotaxis to benzaldehyde? Same for AWA::*cha-1*(RNAi) animals? The current model predicts that they should be.

6) Given the multiple caveats of RNAi, it would be useful to include a control RNAi – so perhaps RNAi against *unc-25* or something like that. If you have such a control, please add it.

7) Given that it is absolutely impossible to know whether the RNAi against *cho-1* was really effective, you should temper their conclusions based on this result.

8) In the present version, you do not discuss whether the effects you report here are specific to the benzaldehyde odorant or whether a similarly distributed circuit also responds to other AWC-sensed odorants. In the previous version, you had mentioned that IAA is different. Please explicitly mention in the Discussion that your BA findings may not be generalizable to all odorants (it is fine to mention the IAA data as unpublished data).

9) In the Discussion, you propose that the “aging-associated sensory impairments are driven by reduced neurotransmitter release from primary neurons”. The data don't really support this conclusion strongly. You show that a) overexpression of *ins-1* as well as b) RNAi of tomosyn results in improved ASE responses and in the latter case, behavior. Similarly, they also show that overexpression of *unc-17* improves responses (also addition of arecoline) and behavior. But these effects need not arise only from reduced release but could also arise from reduced age-dependent expression of cholinergic genes or *ins-1*. Please address this issue in the text.

10) In Figure 5, it would be better to present the scatter plots now included in Figure 5—figure supplement 1. The percentage responsive metric shown in Figure 5 doesn't quite capture the differences in the responses.

[Editors' note: a previous version of this study was rejected after peer review, but the authors submitted for reconsideration. The previous decision letter after peer review is shown below.]

Thank you for choosing to send your work entitled “Neural mechanisms regulating aging-associated behavioral decline in *Caenorhabditis elegans*” for consideration at *eLife*. Your full submission has been evaluated by a Senior Editor, a Reviewing Editor, and three peer reviewers, and the decision was reached after discussions between the reviewers. Based on our discussions and the individual reviews below, we regret to inform you that your work will not be considered further for publication in *eLife*.

While all reviewers recognized the importance and potential interest of the work, the overall sense was that there were too many loose ends in the manuscript that require a substantial amount of additional work. You can see these points listed below by the individual reviews. The member of the Board of Reviewing Editors who handled the manuscript also had concerns about the validity of the claim that the AWA neurons are indeed cholinergic. Once all these concerns have been addressed, you may want to consider submitting the manuscript to *eLife*, but this would count as a completely new submission.

Reviewer #1:

This manuscript by Leinwand and colleagues reports essentially two stories. The first describes the existence of a possible population code for some odors (that were originally thought to be sensed by single sensory neurons), and the second describes the effects of aging on this population coding mechanism, and how changes in this coding may underlie aging-dependent decline in chemotaxis abilities. However, neither story is sufficiently developed.

Essential questions:

If the model is that AWC and AWA represent parallel and non-redundant channels for BZ responses, and that in young animals, AWC and AWA in addition act via ASE and AWB, respectively, why does ASE ablation show as strong a chemotaxis defect as the AWC ablation (1G?). The model would predict that in an ASE-ablated strain, AWC should still be able to drive reasonably strong chemotaxis. This appears to be the case for the AWA/AWB channel, since AWB-ablated animals show a weaker defect than AWA-ablated animals.

Figure 2: The authors assert that AWC is a primary BZ sensor based on the fact that their responses are unaffected in *unc-13* and un*c-31* mutants. There does appear to be a significant defect in the BZ response in *unc-13* mutants though (2A). Also, the responses in *unc-31* mutants are extremely variable. For AWA, the authors can't rule out gap junctions, so their conclusion that AWA is a primary BZ sensor should be tempered. For the rescue experiments (2C, 2D), all of them are quite weak rescue – what is the reason for that? The supplemental data show that ASEL responses are significantly affected in *unc-13* mutants as well but that is not addressed. Finally, it would be useful to see some negative controls here. Easy ones to try are AWA rescue for ASE, and AWC rescue for AWB.

Figure 3: The authors show that ins-1 may be the relevant neuropeptide from AWC that regulates ASE responses to BZ. However, rescue in AWC alone is not sufficient since many reports in *C. elegans* have shown that neuropeptides need not necessarily be expressed from their site of release to have effects. So, the authors need to demonstrate that expression from other neurons do not rescue, or perform RNAi in AWC alone. This is particularly important since *ins-1* released from AIA has previously been shown to affect both AWC responses (by the corresponding author) and ASE responses (Iino group). These latter results should be discussed in the context of this work.

I might have missed this, but are there effects of either *daf-2* or *unc-17* on salt responses in ASE and nonanone responses in AWB, respectively?

While AWB seems to be an OFF neuron for BZ (and also for nonanone, as reported previously), it appears to be an ON neuron for IAA. How might that work? Related to this, the IAA population coding appears to be different than BZ since it looks like AWC shows a pretty strong defect in IAA responses in aged animals. This is shown in the supplemental material but not discussed.

While the data showing that increased NP signaling or cholinergic signaling can rescue the odor response in aged animals are nice, the data leave open the question of exactly how the population code contributes to increased chemotaxis in younger animals. For instance, does the population code change in older animals responses in aversive neurons for instance that compete with neurons driving attraction?

Reviewer #2:

The authors have uncovered a novel sensory circuit motif for the sensation of a medium concentration of benzaldehyde, and have characterized the decline of this ability with age. The authors identified AWC and AWA as primary sensors of this odorant, and ASE and AWB as secondary sensory neurons that are recruited by the primary sensory neurons but do not sense the odor independently. Specifically, they find that the neuropeptide INS-1 is released from AWC and activates ASE through the DAF-2 receptor, while acetylcholine is released from AWA to activate AWB. Age-related behavioral deficits are found to be primarily a consequence of decreased activity in the secondary sensory neurons ASE and AWB in response to the stimulus, but can be corrected by increasing neurotransmission in the primary sensory neurons AWC and AWA. Furthermore, those animals that retained sensory ability with age live longer than animals that did not.

Overall this is a well-written article with novel findings that will contribute meaningfully to circuits/systems neuroscience literature as well as the aging literature. This is particularly timely as the interest in developing additional measures of “healthspan” is being recognized, and cognitive function with age is one aspect that should be studied.

Specific comments and questions:

In the subsection “Insulin Peptidergic and Cholinergic Transmission from Primary Olfactory Sensory Neurons Activates Secondary Olfactory Neurons”: conceptually, if INS-1 is an antagonist how is it used to depolarize ASE?

In the subsection “Attractive Odor-evoked Activity of Secondary Neurons Specifically Decays with Aging”: the authors state that “aging did not affect the reliability, duration, or magnitude of odor-evoked activity” in the primary sensory neurons. However, the heat maps between young and aged neuronal responses appears to show a significant increase in magnitude and duration beyond the 10second window the authors analyze (Figure 4). The same may be said about the odor-responsive aged ASE/AWB being “indistinguishable” from the young.

In the subsection “Attractive Odor-evoked Activity of Secondary Neurons Specifically Decays with Aging”: the authors state that at high concentrations of benzaldehyde the aging-associated activity declines are overcome, however in Figure 4—figure supplement 1 it is clear the behavior has changed significantly. What are their ideas about this? (Some other sensory neuron involved, or issues at the interneuron level…)

In the subsection “Attractive Odor-evoked Activity of Secondary Neurons Specifically Decays with Aging”: the authors state that “declines in neuronal activity might be generally associated with appetitive behavioral deficits in older animals”, but they found repulsive behavioral deficits in older animals as well (Figure 4—figure supplement 1).

In the subsection “Increased Primary Neuron Release Rescues Aging-associated Secondary Neuronal Activity and Behavioral Decay”: no change in the reliability of activity was found in aged animals when increasing DAF-2 receptor expression in ASE, however no control is provided to confirm that receptor expression is sufficiently increased to “young” levels.

In the subsection “Increased Primary Neuron Release Rescues Aging-associated Secondary Neuronal Activity and Behavioral Decay”: while over-expressing INS-1 peptide did improve the reliability of ASE activity in aged animals, it did not rescue the behavior, and even trends toward enhancing the deficit (Figure 5—figure supplement 1).

In the subsection “Increased Primary Neuron Release Rescues Aging-associated Secondary Neuronal Activity and Behavioral Decay”: increasing neurotransmission in AWC and AWA partially rescued behavioral deficits and increased the reliability of AWC and ASE responses. A nice control would be to establish that in these transgenic strains (now overproducing neurotransmitters) the secondary sensory neurons are still necessary for normal behavior (via genetic or laser ablation), especially considering AWA::*unc-17* young worms look as though they may have a significantly higher preference for the odorant than the WT.

In the same section the authors could also report whether acute arecoline treatment improved behavioral deficits.

In the subsection “Olfactory Behavior of Aged Animals is Correlated with Lifespan”: are the lifespans specific to BZ, or general? Testing multiple odorants would establish whether this phenomena is specific to attractive odorants or applies generally to attractive and repulsive odorants.

In the Discussion section: they state that their findings identify function of the secondary (but not primary) neurons decays with age, but in their supplemental materials they also report decay of primary sensory function and associated behavioral deficits to isoamyl alcohol. Others have also identified changes in primary sensory responses (20). So this seems to convolute the claim that “sensory context rather than neuronal identity” is responsible for age-related deficits in sensation.

They suggest that alterations in transmitter release likely underlie age-related cognitive and behavioral decline, which is definitely implied by the experiments they performed, but they did not actually establish that there was a decrease in transmitter release. What was instead established was that increasing transmitter improved behavior (this manipulation could have non-specific effects). These statements could be toned down.

Reviewer #3:

In this manuscript, the authors focused on benzaldehyde (bz) response of amphid sensory neurons and found that in addition to the known bz-sensing neurons AWC, AWA, AWB and ASE respond to bz. AWA shows ON responses and others show OFF responses. According to the authors, AWC and AWA are primary sensory neurons (their response to the odor does not rely on synaptic transmission), and ASE and AWB are secondary sensory neurons. They claim that AWC-to-ASEL communication depends on the insulin signaling genes *ins-1*, *daf-2* and *age-1*, while AWA-to-AWB communication depends on the cholinergic signaling gene *unc-17*. Based on this characterization of the basic circuit, they look at the age-dependent decline of chemosensation. Chemotaxis to bz declines with age, through 1 to 6 days in adulthood. The calcium response of AWC and AWA neurons did not significantly change by age, while response of ASEL and AWB becomes variable in aged animals, whereby fraction of non-responsive neurons increases with age. Increasing the expression of *ins-1* in AWC or AWC-specific RNAi of *tom-1* tomosin rescues the ASEL response, while increasing the expression of *unc-17* or addition of cholinergic agonist arecoline rescues the AWB response. Chemotaxis performance of individual aged animal was found correlated with lifespan. Together, the authors suggest that synaptic transmission is the primary site of age-dependent decline.

This is an interesting paper in which clear changes in bz chemotaxis and underlying sensory circuit during aging was demonstrated and the site of change was identified. This is a beautiful work benefited by the short lifespan and single-cell resolution of the model system of *C. elegans*. The observation that individual animals showing good chemotaxis at day 5 of adulthood are predicted to live longer is particularly interesting.

In spite of these merits, there are also problems in presentation and interpretation of the data as follows.

Major comments:

1) Throughout the manuscript, the authors state that insulin (INS-1) acts for neurotransmission between AWC and ASEL. This terminology itself is probably not wrong, but we need to consider what this statement means in terms of how insulin might work. Are the authors suggesting that it acts as a neurotransmitter in a most classical sense of the term, namely as a factor that is released by excitation of the presynaptic neuron and acts trans-synaptically on the postsynaptic receptor to directly activate the postsynaptic neuron? In this case, how do they think DAF-2 insulin receptor and AGE-1 PI3-kinase might lead to activation of this neuron? Although PI 3-kinase pathway is known to modulate several channels and receptors in mammalian systems, as far as this reviewer knows, adding insulin to a neuron does not immediately excite the neuron.

A more likely possibility is that INS-1 acts chronically on ASEL and modulates the molecular machinery in this neuron so that it becomes more responsive to synaptic inputs to this neuron, for example from AWC. In this case, AWC may use another neurotransmitter such as glutamate to transmit the odor signal to ASEL. As a reference, INS-1 has been reported to act on ASER neuron to modulate this neuron, and DAF-2 receptor in ASER was shown to down-regulate synaptic transmission that occurs in response to chemosensory inputs to this neuron (Ohno et al. Science 2014). It is well established that DAF-16 FOXO acts downstream of DAF-2 to regulate expression of many genes including longevity-related gene. It would therefore be also possible that INS-1 acts hormonally on other nearby neurons to change the expression level of proteins that regulate the activities of this neuron (neurotransmitter receptors, voltage-gated calcium channels etc.). Note that the authors only tested expression of *ins-1* in AWC for rescue but not in other neurons or tissues.

Unless the authors show evidence that INS-1 directly activates ASEL neuron, they should avoid misleading expressions, specifically: “Together, these results indicate that AWC-released insulin peptide signal via the insulin receptor and PI3-Kinase to rapidly activate ASE secondary neurons (within 5 seconds) and encode bz stimulus.” There is no evidence for this statement. Also, the cartoon, Figure 5 middle, clearly implies that INS-1 acts as a classical neurotransmitter. This is also not supported by the results presented.

2) AWA is activated upon addition of bz and AWB is activated upon removal of bz (at the medium concentration). If one assumes that AWB is activated by a direct input from AWA neurons (through acetylcholine), the input ought to be an inhibitory input. Figure 5 right, shows the opposite. Along with the above argument, it is more appropriate to omit Ca^2+^ in ASEL and AWB in the cartoon.

3) The authors often call AWC^ON^ and AWC^OFF^ together as AWC and ASEL and ASER as ASE. However, as stated by the authors in the manuscript these two pairs of neurons are molecularly and functionally different. The authors focused their exploration on AWC^ON^ and ASEL and left out characterization of ASER and AWC^OFF^. Therefore they should be careful in there descriptions. For example, Figure 2–figure supplement 1A shows that calcium response of AWC^OFF^ is significantly compromised in the *unc-13* mutant even if one looks only at the initial increase in the calcium level. Therefore AWC^OFF^ is categorized as secondary olfactory neuron for bz, while the authors state “two pairs of primary sensory neurons (AWC and AWA)”. This is inappropriate.

4) In the subsection “Attractive Odor-evoked Activity of Seconardy Neurons Specifically Decays with Aging: “the calcium transient of odor-responsive aged ASEL and AWB neurons were indistinguishable from responses in younger animals”. I actually do not see this. If we leave out non-responsive animals, by manual inspection, overall response of ASEL looks higher and that of AWB looks lower. The authors need to quantify the magnitude of response of “responding” neurons. This is important because the authors claim that variability of neuronal responses increases by age. This is true if the magnitude of the calcium response of “responding” neurons in old animals is equal or higher than young animals, because there are more “non-responsive” animals, while if response of responding neurons are lower in old animals than in young animals, this simply means an overall decline of neuronal response.

5) To me, the results of Figure 4–figure supplement 3 look contradictory with the results in Figure 4. The former indicate that the sensory responses of ASEL and AWB are correlated with chemotaxis performance, while the latter indicates that the presence of ASEL or AWB neurons are irrelevant to chemotaxis performance in aged animals. One possible explanation to solve this contradiction would be that responsiveness of ASEL and AWB is correlated with responsiveness (or other functions) of AWC or AWA, so that an individual with poor response of ASEL has a poor function of AWC^ON^. Is this the case?

---

## [Author Response]

*1) AWB shows a nice odor-off response even in AWA-ablated animals. The source of this sensory stimulus is not known, one possibility being redundant inputs from AWA and AWC. Small but significant AWB response is also observed in the* unc-13 *mutant,* unc-17 *mutant and other manipulations. Given the status of the analysis and the characteristics of AWB as olfactory neurons, direct sensory input to AWB is not excluded and is in fact likely. Please discuss this issue in the text and put an arrow (maybe dotted arrow) from “benzaldehyde” to “AWB” in*
Figures 2 and 4*, unless you identify neurons that account for 100% of the odor-off response of AWB*.

We agree with the reviewers that our results do not exclude the possibility that AWB neurons receive some direct sensory input. We have added some subsections to clarify this point by adding to our descriptions of the cell ablation results and the *unc-13* and AWA neuron-specific tetanus toxin expression results (subsections “Primary and Secondary Olfactory Neurons Encode Benzaldehyde Odor” and “AWC-released Neuropeptides and AWA-released Classical Neurotransmitters are Required for the Activity of ASEL and AWB Neurons, Respectively”). We have also revised Figures 2 and 4 by drawing dotted arrows in our models to reflect this possibility.

*2) Also, AWA's role is enigmatic. Upon odor removal, at which time AWB is activated, AWA does not respond at all, or show only a small calcium increase. On the contrary, upon odor addition, AWA is largely activated but AWB is not activated, or a bit inhibited. Therefore a simple excitatory or inhibitory synapse from AWA to AWB does not explain the behavior of AWB neurons. This simple consideration leads to the question whether AWBs dependence on AWA is based on fast synaptic transmission. You show that the* unc-17 *mutant shows reduced response of AWB and this defect is rescued by* unc-17 *expression in AWA.* cho-1 *RNAi shows similar results. However, this does not necessarily mean AWB is activated upon odor removal by cholinergic neurotransmission from AWA. Equally possible explanations – which you should discuss in the text – are the modulatory effects of AWA-released acetylcholine (as muscarinic receptors function in other systems), by which AWB becomes more responsive to direct odor sensation. In the Abstract, you state: “Primary neurons, AWC*^*ON*^
*and AWA, directly detect the food odor benzaldehyde and release insulin-like peptides and acetylcholine, respectively, to activate secondary neurons, ASEL and AWB”. This clearly implicates that insulin and acetylcholine are released upon odor detection to activate ASEL and AWB. This is not supported because there is no release timing data presented for neither insulin nor acetylcholine (these data are generally difficult to obtain in* C. elegans*). You need to carefully rephrase these statements to something like “insulin-like peptide and acetylcholine released from AWC and AWA respectively are required for activity of secondary neurons”*.

We rephrased our statements about the effects of primary olfactory neuron released insulin-like peptides and acetylcholine in the Abstract and in the section headings to better describe our results. We also expanded this discussion to describe possible modulatory effects of AWA-released acetylcholine to sensitize AWB responses to odor (please see Results section description of Figure 3). However, we consider modulatory signaling through muscarinic acetylcholine receptors unlikely because we observed wild-type like AWB odor responses in our experiments examining mutations in the only three identified *C. elegans* muscarinic type receptors (*gar-1, gar-2* and *gar-3*).

*3) The quantification data in*
Figure 5—figure supplement 1
*are persuasive in showing that only ASEL and AWB decline by age. However, you need to accurately describe the quantification method. In the Methods, you state: “To analyze off responses to the removal of stimulus, F0 was set to the time just prior to odor removal (t=121-129s)”. This is probably not what you did for the heat maps in*
Figure 5*,*
Figure 6*,*
Figure 7*, which show only the 120s-180s time points, and the dot plots in*
Figure 5—figure supplement 1*. Instead, F0 seems to come from the average of t=1-9s for these analyses*.

We made several changes to clarify our quantification methods. First, for all the truncated heat maps shown in Figures 5, 6 and 7, we changed the time labels (on the x-axis) to show 120-180s, which more accurately describes the part of the experiment depicted and allows us to refer to particular time windows in our imaging experiments in one consistent manner throughout the entire paper. (We note that we did not change the time labels for Figure 5, which shows 0-60s and includes the odor addition period in which AWA neurons are active.) Next, we revised our Materials and methods section text to clearly state the F0 used for each heat map or graph. While the Calcium Imaging subsection of the Materials and methods section now elaborates on the analysis used in Figures 1, 2, 3, 4, 5, 6 and 7, here (Table 1) we describe only the protocol as it pertains to the reviewers' questions about Figures 5, 6 and 7.

Heat maps: In keeping with these time labels, we now state explicitly in the Methods that the F0 used to generate all of our heat maps is the average of t=1-9s.

Quantifications: F0 is consistently set to the average of the signal from 1 to 9s just prior to the relevant change (addition or removal) of stimulus. Specifically:

A) For the scatter plot of Maximum ΔF/F (Figure 5) and the scatter plots of Averaged ΔF/F after stimulus change (Figure 5, Figure 6—figure supplement 1 and Figure 7—figure supplement 1): (a) F0=1-9s for AWA neurons' response to the addition of odor stimulus and (b) F0=121-129s for AWC^ON^, ASEL and AWB responses to odor stimulus removal.

B) For our bar graph quantifications of the % odor (or salt) responsive (Figures 5, 6 and 7 and the corresponding figure supplements): (a) F0=1-9s for quantification of the percent of AWA and ASH neurons responsive to the addition of benzaldehyde stimulus and for the percent of ASEL and AWC neurons responsive to the addition of NaCl salt stimulus. (b) F0=121-129s for AWC^ON^, ASEL and AWB responses to benzaldehyde or nonanone odor stimulus removal.

*4) In the Methods, you state: “the percent of odor responsive neurons was calculated by determining the proportion of cells displaying an average fluorescence (ΔF/F) greater than 10 percent after odor addition (for AWA) or odor removal (all other neurons)”. For*
Figure 5, Figure 6*,*
Figure 7
*and other similar figures, which time points did you use for calculating F0 (or F), 1-9s or 121-129s?*

*The definition of F0, of course, affects the interpretation of the results. All time series traces of ASEL and AWB calcium levels, like those in*
Figure 2
*show downward trends during odor exposure. If F0 is defined for 1-9s, all data quoted above might mean that aged ASEL and AWB are more sensitive to suppression by sustained odor exposure. This can be appreciated by quantifying the decline in calcium levels during 2-minutes odor exposure in young and aged animals, but this result is not presented and it should be.*

*It is fine either way, as far as there is a change in “secondary neuron” responses between young and aged animals, but the results must be accurately presented so that readers are not lead to misunderstandings*.

We have revised our Calcium imaging subsection of the Materials and methods section to clarify our analysis protocol, as described above in response to #3. For the quantifications of young and aged ASEL and AWB odor responses, we have set F0 as the average fluorescence signal from 1s to 9s immediately prior to the stimulus removal (t=121-129s). For many cells, this quantification method reveals larger magnitude responses than the responses that would have been calculated if F0 were set to t=1-9s; however, we chose this method because it is more sensitive to small odor-evoked increases in the calcium signal. Crucially, we note that, considering only the odor responsive neurons, we did not observe any significant differences between in the magnitude of the odor-evoked suppression of young and aged animals (please see Table 1). This indicates that our choice of F0 for the analysis of the removal of odor is unlikely to bias our conclusions about aging-associated declines in neuronal activity.

Statistical analysis of odor-evoked suppression summary chart comparing young and aged adult wild-type sensory neuron calcium responses to medium benzaldehyde:

Author response table 1.Comparison of Averaged ΔF/F in 10s time window listed below for young and aged WT odor responsive neurons (by two-tailed *t*-test)**DOI:**
http://dx.doi.org/10.7554/eLife.10181.046Neuron10-20s20-30s30-40s40-50s50-60s60-70s**AWC**^ON^NS, P=0.1682NS, P=0.2114NS, P=0.2306NS, P=0.3829NS, P=0.4481NS, P=0.5167**ASEL**NS, P=0.2593NS, P=0.1983NS, P=0.1421NS, P=0.2748NS, P=0.2067NS, P=0.1822**AWB**NS, P=0.0601NS, P=0.1168NS, P=0.1327NS, P=0.2933NS, P=0.3453NS, P=0.3298**Neuron****70-80s****80-90s****90-100s****100-110s****110-120s****120-129s****AWC**^ON^NS, P=0.4226NS, P=0.4927NS, P=0.4857NS, P=0.5729NS, P=0.5361NS, P=0.6571**ASEL**NS, P=0.1531NS, P=0.1408NS, P=0.1446NS, P=0.1471NS, P=0.0620NS, P=0.0895**AWB**NS, P=0.2249NS, P=0.4059NS, P=0.2927NS, P=0.3530NS, P=0.2368NS, P=0.2538

(Note: This analysis was performed only on the subset of odor responsive neurons. F0 was set to the average fluorescence from t=1-9s for this analysis of the odor-evoked suppression.)

*5) Please indicate whether you tested whether* ins-1 *mutants are defective in chemotaxis to benzaldehyde? Same for AWA::*cha-1*(RNAi) animals? The current model predicts that they should be.*

We have added new data showing that both *AWC::*ins-*1 RNAi* and *AWA::*cha-1 *RNAi* animals exhibit defective chemotaxis to a medium concentration of benzaldehyde (see Figure 4).

*6) Given the multiple caveats of RNAi, it would be useful to include a control RNAi – so perhaps RNAi against* unc-25 *or something like that. If you have such a control, please add it*.

Our experimental approach typically began with the analysis of genetic mutants and cell-specific rescue experiments (for example, we found that AWB neuron responses to benzaldehyde were unchanged in *eat-4, tph-1* and *cat-2* mutants compared to wild-type); consequently, we do not have additional negative control RNAi data to include.

We also refer the reviewers to our previously published paper (40) where we used the same RNAi knockdown approach and tested the specificity of this approach with negative control experiments targeting different neurotransmission pathway genes and also targeting alternate amphid sensory neurons (using alternate cell-specific promoters).

*7) Given that it is absolutely impossible to know whether the RNAi against* cho-1 *was really effective, you should temper their conclusions based on this result.*

We agree with the reviewers that it is not feasible to measure the effect of RNAi in a single pair of *C. elegans* neurons; therefore, we mention this and temper our conclusions about the role of *cho-1* in the Results section (subsection “Insulin Peptidergic and Cholinergic Transmission from Primary Olfactory Sensory Neurons are Required for Secondary Olfactory Neuron Activity”).

8) In the present version, you do not discuss whether the effects you report here are specific to the benzaldehyde odorant or whether a similarly distributed circuit also responds to other AWC-sensed odorants. In the previous version, you had mentioned that IAA is different. Please explicitly mention in the Discussion that your BA findings may not be generalizable to all odorants (it is fine to mention the IAA data as unpublished data).

We now state in the Discussion that a distinct, but similarly distributed neural circuit (which does not include the ASE neurons) encodes a different attractive odorant, isoamyl alcohol.

*9) In the Discussion, you propose that the “aging-associated sensory impairments are driven by reduced neurotransmitter release from primary neurons”. The data don't really support this conclusion strongly. You show that a) overexpression of* ins-1 *as well as b) RNAi of tomosyn results in improved ASE responses and in the latter case, behavior. Similarly, they also show that overexpression of* unc-17 *improves responses (also addition of arecoline) and behavior. But these effects need not arise only from reduced release but could also arise from reduced age-dependent expression of cholinergic genes or* ins-1*. Please address this issue in the text.*

Our experiments show that experimental manipulations aimed at increasing neurotransmitter release improve the aging-associated neuronal activity and behavioral declines. Furthermore, we used quantitative RT-PCR to measure the relative mRNA levels of *ins-1, daf-2* and *unc-17* in young (Day 1) and aged (Day 5) adults and did not observe any significant differences in expression levels (data from three biological replicates is included here for the reviewers). This is consistent with data from Jin et al. (Cell Metab 2011) showing no change (or a small increase) in expression of many canonical insulin signaling pathway genes in aging *C. elegans.* This data suggests that reduced expression of insulin and cholinergic pathway genes in aging animals may not be the driver of aging-associated olfactory declines. However, this experiment was performed with RNA from the entire animal; therefore, specific changes in gene expression within the small number of neurons that comprise the olfactory circuit cannot be ruled out. We have revised our manuscript to discuss this and the different possible mechanisms that could underlie the declines observed in aged animals and which could be overcome by our manipulations (Discussion).

*10) In*
Figure 5*, it would be better to present the scatter plots now included in*
Figure 5—figure supplement 1*. The percentage responsive metric shown in*
Figure 5
*doesn't quite capture the differences in the responses.*

We rearranged Figure 5 and Figure 5—figure supplement 1 such that the scatter plots showing the maximum ΔF/F and the averaged ΔF/F after stimulus change for young and aged adult responses to benzaldehyde are now included in the main Figure 5.

[Editors' note: the author responses to the previous round of peer review follow.]

We have made substantial revisions to our manuscript and changed the title to “Circuit mechanisms encoding odors and driving aging-associated behavioral declines in *Caenorhabditis elegans*” in response to the comments from the reviewers and from a member of the Board of Reviewing Editors. The largest change we made was the addition of data combining cell ablations with calcium imaging to confirm the primary and secondary neuron functions that we had previously defined using genetics. We also used cell-specific RNAi knockdown experiments to further support our conclusions that (1) AWA primary olfactory neurons release acetylcholine and that (2) AWC olfactory neurons release INS-1 to recruit secondary neurons in the olfactory circuit. We have also included additional quantifications and experiments to further support our conclusions about aging-associated declines and the correlation between olfactory circuit function and lifespan. Finally, we have also reorganized our paper to discuss our combinatorial coding strategy before describing how this code is changed during the aging process.

*While all reviewers recognized the importance and potential interest of the work, the overall sense was that there were too many loose ends in the manuscript that require a substantial amount of additional work. You can see these points listed below by the individual reviews. The member of the Board of Reviewing Editors who handled the manuscript also had concerns about the validity of the claim that the AWA neurons are indeed cholinergic. Once all these concerns have been addressed, you may want to consider submitting the manuscript to* eLife*, but this would count as a completely new submission.*

We used cell-specific knockdown experiments to further support our conclusion that AWA neurons can release acetylcholine (Figure 4). We find that knocking down *cha-1* (the *C. elegans* homolog of the choline acetyltransferase which is required for the biosynthesis of acetylcholine) specifically in AWA cells blocks AWB responses to benzaldehyde. These results along with our AWA neuron specific rescue of *unc-17* (the vesicular acetylcholine transporter) provide two lines of evidence showing that AWA neurons can release acetylcholine in response to benzaldehyde stimulus to recruit AWB neurons into the olfactory neural circuit.

Our experiments to knockdown *cho-1* (the choline reuptake transporter) in AWA neurons had no effect on AWB responses to benzaldehyde; however, this result is not incompatible with AWA neurons being cholinergic as previous studies have shown that loss of *cho-1* has only mild effects on cholinergic neurotransmission (46).

Reviewer #1:

This manuscript by Leinwand and colleagues reports essentially two stories. The first describes the existence of a possible population code for some odors (that were originally thought to be sensed by single sensory neurons), and the second describes the effects of aging on this population coding mechanism, and how changes in this coding may underlie aging-dependent decline in chemotaxis abilities. However, neither story is sufficiently developed.

We recognize the reviewer's concerns about the two stories in our manuscript. We have reorganized our manuscript to first describe the combinatorial coding strategy used by *C. elegans* chemosensory circuits and then discuss how this code is changed during the aging process.

Essential questions:

If the model is that AWC and AWA represent parallel and non-redundant channels for BZ responses, and that in young animals, AWC and AWA in addition act via ASE and AWB, respectively, why does ASE ablation show as strong a chemotaxis defect as the AWC ablation (1G?). The model would predict that in an ASE-ablated strain, AWC should still be able to drive reasonably strong chemotaxis. This appears to be the case for the AWA/AWB channel, since AWB-ablated animals show a weaker defect than AWA-ablated animals.

The reviewer is correct in identifying the two parallel channels (AWC-ASE and AWA-AWB) for benzaldehyde responses. There is a difference in signaling between these two channels in that AWA might signal through multiple downstream cells including AWB, while AWC might have fewer downstream targets including ASE. It is therefore not surprising that ASE (-) animals have a similar phenotype as AWC (-) animals.

Figure 2*: The authors assert that AWC is a primary BZ sensor based on the fact that their responses are unaffected in* unc-13 *and* unc-31 *mutants. There does appear to be a significant defect in the BZ response in* unc-13 *mutants though (2A). Also, the responses in* unc-31 *mutants are extremely variable. For AWA, the authors can't rule out gap junctions, so their conclusion that AWA is a primary BZ sensor should be tempered. For the rescue experiments (2C, 2D), all of them are quite weak rescue – what is the reason for that? The supplemental data show that ASEL responses are significantly affected in* unc-13 *mutants as well but that is not addressed. Finally, it would be useful to see some negative controls here. Easy ones to try are AWA rescue for ASE, and AWC rescue for AWB.*

We have added an independent method to confirm the primary and secondary neurons in the benzaldehyde circuit. We combined cell ablations with calcium imaging, predicting that primary neuron responses would not be affected by the ablation of other sensory cells, while secondary neuron responses would be greatly reduced when primary neurons are ablated. These experiments confirm that AWC^ON^ and AWA are primary neurons, while ASEL and AWB are secondary neurons. These results are also consistent with our genetic experiments. Moreover, we have re-analyzed our data and find that ASEL responses in *unc-13* mutants are not significantly different from those in WT animals (p = 0.098 two-tailed *t*-test). (In this analysis and all the experiments in the paper we have now excluded traces in which an averaged ΔF/F of greater than 600% was recorded after the stimulus change. Such traces account for less than 1% of the traces collected and are likely to be artifacts of animal movement causing the cell to move out of the focal plane etc.) The reviewer is correct in identifying that AWC^OFF^ neuron is not a primary neuron for benzaldehyde as its odor responses are significantly reduced in *unc-13* mutants. We have edited our revised version to reflect that.

Figure 3*: The authors show that* ins-1 *may be the relevant neuropeptide from AWC that regulates ASE responses to BZ. However, rescue in AWC alone is not sufficient since many reports in* C. elegans *have shown that neuropeptides need not necessarily be expressed from their site of release to have effects. So, the authors need to demonstrate that expression from other neurons do not rescue, or perform RNAi in AWC alone. This is particularly important since* ins-1 *released from AIA has previously been shown to affect both AWC responses (by the corresponding author) and ASE responses (Iino group). These latter results should be discussed in the context of this work*.

We thank the reviewer for this valuable suggestion. We have performed AWC cell-specific RNAi for *ins-1* and find that this manipulation reduces ASEL responses to benzaldehyde. We also show that rescuing *ins-1* in AWA has no effect on ASEL responses. We have included this data in Figure 4 and also discussed these results along with previously described studies.

*I might have missed this, but are there effects of either* daf-2 *or* unc-17 *on salt responses in ASE and nonanone responses in AWB, respectively?*

We have previously shown that ASE responses to salt are not affected in *daf-2* or *age-1* mutants (40). We have also included data to show that AWB responses to 2-nonanone are not significantly affected in the *unc-17* mutants (Figure 4—figure supplement 1).

*While AWB seems to be an OFF neuron for BZ (and also for nonanone, as reported previously), it appears to be an ON neuron for IAA. How might that work? Related to this, the IAA population coding appears to be different than BZ since it looks like AWC shows a pretty strong defect in IAA responses in aged animals. This is shown in the supplemental material but not discussed*.

We have streamlined our manuscript to focus on the benzaldehyde circuit. Our lab continues to probe these olfactory circuits and we feel that the isoamyl circuit is outside the scope of this work.

While the data showing that increased NP signaling or cholinergic signaling can rescue the odor response in aged animals are nice, the data leave open the question of exactly how the population code contributes to increased chemotaxis in younger animals. For instance, does the population code change in older animals responses in aversive neurons for instance that compete with neurons driving attraction?

We suggest that the combinatorial code allows younger animals to respond robustly to odor gradients. We have analyzed the neural code of an aversive concentration of benzaldehyde and find that it is not changed in older animals (Figure 5—figure supplement 3). We suggest that the neural code for volatile attractants might be more susceptible to the deleterious effects of aging when compared to the neural code for repellents under these assay conditions. Broadly speaking, we do agree with the reviewer that the aging-associated change in the combinatorial code is likely to influence the neural circuit at the level of downstream interneurons.

*Reviewer #2*:

*In the subsection “Insulin Peptidergic and Cholinergic Transmission from Primary Olfactory Sensory Neurons Activates Secondary Olfactory Neurons”*: *conceptually, if INS-1 is an antagonist how is it used to depolarize ASE?*

We are not sure how or if INS-1 depolarizes ASE neurons. We have edited our manuscript to say that INS-1 recruits ASE into the benzaldehyde circuit. While the canonical effects of the DAF-2/AGE-1 pathway regulate transcription and gene expression (described by Murphy CT et al. and others), we speculate that alternate pathways also exist which may be homologous to the rapid effect of insulin-like growth factor-1 (IGF-1) to increase calcium channel currents within seconds (dependent on receptor tyrosine kinase and PI3-Kinase signaling) (12).

In this way, the increase in calcium in ASEL may result from rapid signaling downstream of INS-1, DAF-2 and PI3-Kinase.

*In the subsection “Attractive Odor-evoked Activity of Secondary Neurons Specifically Decays with Aging”: the authors state that “aging did not affect the reliability, duration, or magnitude of odor-evoked activity” in the primary sensory neurons. However, the heat maps between young and aged neuronal responses appears to show a significant increase in magnitude and duration beyond the 10second window the authors analyze (*Figure 4*). The same may be said about the odor-responsive aged ASE/AWB being “indistinguishable” from the young*.

We include two plots in our supplementary data to show that the reliability, duration or magnitude of the primary neurons is not affected between young and aged animals (Figure 5—figure supplement 1). These graphs also show that while the reliability of AWB secondary is reduced in aged animals, the average size of the AWB response and the maximum (peak) response are unchanged by aging (Figure 5—figure supplement 1). Considering only the odor-responsive ASEL neurons, our analysis finds that the ASEL response to benzaldehyde removal is slightly but significantly larger in aged animals (Figure 5—figure supplement 1); we have amended the text accordingly. We have also compared our data in other 10s time windows beyond the time period immediately after the stimulus addition/removal and do not see any significant difference (see Table 2).

Author response table 2.Is there a significant difference between the young and the aged odor responsive neurons?**DOI:**
http://dx.doi.org/10.7554/eLife.10181.047Neuron130-140s140-150s150-160s160-170s170-180s**AWC**^ON^NS, P=0.0643NS, P=0.1383NS, P=0.6332NS, P=0.1942NS, P=0.0939**ASEL*****P=0.0336**NS, P=0.0599NS, P=0.1584NS, P=0.4384NS, P=0.4712**AWB**NS, P=0.0909NS, P=0.0709NS, P=0.0886NS, P=0.3076NS, P=0.5934**Neuron****10-20s****20-30s****30-40s****40-50s****50-60s****AWA**NS, P=0.3929NS, P=0.3771NS, P=0.1101NS, P=0.1425NS, P=0.1957

*In the subsection “Attractive Odor-evoked Activity of Secondary Neurons Specifically Decays with Aging”: the authors state that at high concentrations of benzaldehyde the aging-associated activity declines are overcome, however in*
Figure 4—figure supplement 1
*it is clear the behavior has changed significantly. What are their ideas about this? (Some other sensory neuron involved*, *or issues at the interneuron level…)*

Animals avoid high concentrations of benzaldehyde. Through Day 5 of adulthood (the early stage of aging examined throughout), we do not observe a significant change in chemotaxis to high benzaldehyde compared to young adults (Figure 5—figure supplement 3). Our imaging data shows that a combinatorial circuit encodes this repulsive concentration, but this code does not change in the early stage of aging. We suggest that attractive circuits might degrade earlier when compared to repulsive circuits during aging.

*In the subsection “Attractive Odor-evoked Activity of Secondary Neurons Specifically Decays with Aging”: the authors state that “declines in neuronal activity might be generally associated with appetitive behavioral deficits in older animals”, but they found repulsive behavioral deficits in older animals as well (*Figure 4—figure supplement 1*)*.

We did observe repulsive behavioral deficits in older Day 6 animals but not in the Day 5 aged animals. Our imaging experiments were all done in Day 5 adults. We have removed the older Day 6 data to avoid confusion as it was not discussed and was not central to the conclusions presented in this manuscript.

*In the subsection “Increased Primary Neuron Release Rescues Aging-associated Secondary Neuronal Activity and Behavioral Decay”: no change in the reliability of activity was found in aged animals when increasing DAF-2 receptor expression in ASE, however no control is provided to confirm that receptor expression is sufficiently increased to “young” levels*.

It is difficult to accurately measure DAF-2 expression levels in extrachromosomal transgenic animals. However, we analyzed the neuronal activity of DAF-2 over-expressing animals in younger animals and found that they had stronger ASE responses (Figure 6—figure supplement 1). These results show that the DAF-2 array is functional, but does not have an effect on improving aged ASE responses to benzaldehyde.

*In the subsection “Increased Primary Neuron Release Rescues Aging-associated Secondary Neuronal Activity and Behavioral Decay”: while over-expressing INS-1 peptide did improve the reliability of ASE activity in aged animals, it did not rescue the behavior, and even trends toward enhancing the deficit (*Figure 5—figure supplement 1*)*.

We are equally puzzled by this result. We suggest that INS-1 is likely to have multiple functions and perhaps over-expressing it from AWC affects many pathways with opposing effects on chemotaxis behavior irrespective of the animal age. We clarify the text in this regard.

*In the subsection “Increased Primary Neuron Release Rescues Aging-associated Secondary Neuronal Activity and Behavioral Decay”: increasing neurotransmission in AWC and AWA partially rescued behavioral deficits and increased the reliability of AWC and ASE responses. A nice control would be to establish that in these transgenic strains (now overproducing neurotransmitters) the secondary sensory neurons are still necessary for normal behavior (via genetic or laser ablation), especially considering AWA::*unc-17 *young worms look as though they may have a significantly higher preference for the odorant than the WT*.

We thank the reviewer for this suggestion. We include additional data to show that increased neurotransmission from AWC and AWA neurons requires ASE and AWB secondary neurons for the rescue of behavioral defects in aged animals (Figures 6 and 7).

*In the same section the authors could also report whether acute arecoline treatment improved behavioral deficits*.

We have attempted the acute arecoline treatment prior to behavioral assays. We find that the locomotion of the drug-treated animals is altered and the chemotaxis of young animals is significantly reduced by this treatment (data included in Figure 7—figure supplement 1. We suggest that the known effect of arecoline to increase spontaneous locomotion may be counterproductive to the directed locomotion required to chemotax up an odor gradient

*In the subsection “Olfactory Behavior of Aged Animals is Correlated with Lifespan”: are the lifespans specific to BZ, or general? Testing multiple odorants would establish whether this phenomena is specific to attractive odorants or applies generally to attractive and repulsive odorants*.

We include additional data to show that aged animals that successfully chemotax to salt have similar lifespans to those that fail to chemotax to salt. We feel that other odorants are beyond the scope of this study.

*In the Discussion section: they state that their findings identify function of the secondary (but not primary) neurons decays with age, but in their supplemental materials they also report decay of primary sensory function and associated behavioral deficits to isoamyl alcohol. Others have also identified changes in primary sensory responses (*[20]*). So this seems to convolute the claim that “sensory context rather than neuronal identity” is responsible for age-related deficits in sensation*.

We find that the decay of secondary neuron functions during aging is highly specific. For example, ASE responses to salt are preserved, while ASE responses to benzaldehyde decline. Moreover, not all secondary responses decay with age. e.g. AWC secondary responses to salt do not decay with age. These data suggest that sensory context and neuronal identity together (not neuronal identity alone) determine whether there is an age-related decay. We clarified our discussion about these results and the results of Chokshi et al. in our Discussion section.

*They suggest that alterations in transmitter release likely underlie age-related cognitive and behavioral decline, which is definitely implied by the experiments they performed, but they did not actually establish that there was a decrease in transmitter release. What was instead established was that increasing transmitter improved behavior (this manipulation could have non-specific effects). These statements could be toned down*.

We have edited the manuscript and toned down our statements in this regard.

*Reviewer #3*:

*Major comments*:

*1) […] Unless the authors show evidence that INS-1 directly activates ASEL neuron, they should avoid misleading expressions, specifically: “Together, these results indicate that AWC-released insulin peptide signal via the insulin receptor and PI3-Kinase to rapidly activate ASE secondary neurons (within 5 seconds) and encode bz stimulus.” There is no evidence for this statement. Also, the cartoon,*
Figure 5
*middle, clearly implies that INS-1 acts as a classical neurotransmitter. This is also not supported by the results presented*.

We agree with the reviewer and do not have a direct method to show that AWC released INS-1 activates ASE. We have included additional data showing that AWC neuron specific knockdown of *ins-1* impairs ASE odor responses, while AWA-neuron specific rescue of *ins-1* fails to rescue the mutant odor responses. Additionally, we agree with the reviewer that the canonical effects of the DAF-2/AGE-1 pathway regulate transcription and gene expression (described by Murphy CT et al. and others) and we cannot rule out that this leads to a chronic action of INS-1 to modulate the molecular machinery of the ASEL neuron. However, we speculate that alternate pathways downstream of INS-1/DAF-2 could also exist to rapidly modulate neuronal activity. These alternate pathways may be homologous to the rapid effect of insulin-like growth factor-1 (IGF-1) to increase calcium channel currents within seconds to alter neuronal activity (dependent on receptor tyrosine kinase and PI3-Kinase signaling) ([12], Selinfreund and Blair 1994). In this way, the increase in calcium in ASEL might result from rapid signaling downstream of INS-1, DAF-2 and PI3-Kinase acting directly on calcium channels. Because this is speculative, we have edited our manuscript to say that simply “AWC released INS-1 recruits ASE into the benzaldehyde circuit”. We have also changed our cartoons throughout the paper.

*2) AWA is activated upon addition of bz and AWB is activated upon removal of bz (at the medium concentration). If one assumes that AWB is activated by a direct input from AWA neurons (through acetylcholine), the input ought to be an inhibitory input.*
Figure 5
*right, shows the opposite. Along with the above argument, it is more appropriate to omit Ca*^*2+*^
*in ASEL and AWB in the cartoon*.

We agree with the reviewer and changed our cartoon to reflect this concern.

*3) The authors often call AWC*^*ON*^
*and AWC*^*OFF*^
*together as AWC and ASEL and ASER as ASE. However, as stated by the authors in the manuscript these two pairs of neurons are molecularly and functionally different. The authors focused their exploration on AWC*^*ON*^
*and ASEL and left out characterization of ASER and AWC*^*OFF*^*. Therefore they should be careful in there descriptions. For example, Figure 2–figure supplement 1A shows that calcium response of AWC*^*OFF*^
*is significantly compromised in the* unc-13 *mutant even if one looks only at the initial increase in the calcium level. Therefore AWC*^*OFF*^
*is categorized as secondary olfactory neuron for bz, while the authors state “two pairs of primary sensory neurons (AWC and AWA)”. This is inappropriate*.

We agree with the reviewer and have edited our manuscript. We find that AWC^ON^ and AWA are primary neurons, while ASEL and AWB are secondary neurons.

*4) In the subsection “Attractive Odor-evoked Activity of Seconardy Neurons Specifically Decays with Aging: “the calcium transient of odor-responsive aged ASEL and AWB neurons were indistinguishable from responses in younger animals”. I actually do not see this. If we leave out non-responsive animals, by manual inspection, overall response of ASEL looks higher and that of AWB looks lower. The authors need to quantify the magnitude of response of “responding” neurons. This is important because the authors claim that variability of neuronal responses increases by age. This is true if the magnitude of the calcium response of “responding” neurons in old animals is equal or higher than young animals, because there are more “non-responsive” animals, while if response of responding neurons are lower in old animals than in young animals, this simply means an overall decline of neuronal response*.

We have included additional data in Figure 5—figure supplement 1 quantifying the neuronal responses. Considering only the animals with odor-responsive ASEL neurons, our analysis finds that the ASEL response to benzaldehyde removal is significantly larger in aged animals than in young animals (Figure 5—figure supplement 1). For the animals with odor-responsive AWB neurons, the average size of the AWB response and the peak response are unchanged by aging (Figure 5—figure supplement 1). We have amended the text accordingly.

*5) To me, the results of Figure 4-figure supplement 3 look contradictory with the results in*
Figure 4*. The former indicate that the sensory responses of ASEL and AWB are correlated with chemotaxis performance, while the latter indicates that the presence of ASEL or AWB neurons are irrelevant to chemotaxis performance in aged animals. One possible explanation to solve this contradiction would be that responsiveness of ASEL and AWB is correlated with responsiveness (or other functions) of AWC or AWA, so that an individual with poor response of ASEL has a poor function of AWC*^*ON*^*. Is this the case?*

We find that AWC^ON^ responses are reliable in aged animals, while ASEL responses are reduced and unreliable (quantified in Figure 5—figure supplement 1). We do see a correlation with chemotaxis performance and odor responsiveness of secondary neurons (Figure 5–Figure 5—figure supplement 2). We show that the improved chemotaxis performance observed in animals with increased neurotransmission from primary neurons requires functional secondary neurons (Figures 6 and 7). Together these results show that the secondary neurons play a crucial role in aged animal chemotaxis.